# Randomized Sparse Neural Galerkin Schemes for Solving Evolution Equations with Deep Networks

**Jules Berman**
Courant Institute for Mathematical Sciences
New York University
New York, NY 10012
jmb1174@nyu.edu

**Benjamin Peherstorfer**
Courant Institute for Mathematical Sciences
New York University
New York, NY 10012
pehersto@cims.nyu.edu

## Abstract

Training neural networks sequentially in time to approximate solution fields of time-dependent partial differential equations can be beneficial for preserving causality and other physics properties; however, the sequential-in-time training is numerically challenging because training errors quickly accumulate and amplify over time. This work introduces Neural Galerkin schemes that update randomized sparse subsets of network parameters at each time step. The randomization avoids overfitting locally in time and so helps prevent the error from accumulating quickly over the sequential-in-time training, which is motivated by dropout that addresses a similar issue of overfitting due to neuron co-adaptation. The sparsity of the update reduces the computational costs of training without losing expressiveness because many of the network parameters are redundant locally at each time step. In numerical experiments with a wide range of evolution equations, the proposed scheme with randomized sparse updates is up to two orders of magnitude more accurate at a fixed computational budget and up to two orders of magnitude faster at a fixed accuracy than schemes with dense updates.

## 1 Introduction

In science and engineering, partial differential equations (PDEs) are frequently employed to model the behavior of systems of interest. For many PDEs that model complicated processes, an analytic solution remains elusive and so computational techniques are required to compute numerical solutions.

**Global-in-time training**   There have been many developments in using nonlinear parameterizations based on neural networks for numerically approximating PDE solutions. These include techniques such as the Deep Galerkin Method [46], physics-informed neural networks (PINNs) [41], and others [4, 21, 53, 15]; as well as early works such as [11, 42]. In most of these methods, a neural network is used to represent the solution of a (time-dependent) PDE over the whole space-time domain. For this reason they are termed global-in-time methods in the following. To approximate the solution, the neural network is trained to minimize the PDE residual on collocation points sampled from the space-time domain, which requires solving a large-scale optimization problem that can be computationally expensive. Additionally, the solutions learned by global-in-time methods can violate causality, which can become an issue for complex problems that rely on preserving physics [27]. We note that neural networks have been used for approximating PDE solutions in various other ways, such as learning specific component functions [23, 31, 43], finding closure models [2, 25, 50], de-noising [44], and for surrogate modeling [33, 32, 18]. However, we are interested in this work in using neural networks for directly approximating PDE solutions.

37th Conference on Neural Information Processing Systems (NeurIPS 2023).

**Sequential-in-time training with the Dirac-Frenkel variational principle** In this work, we follow the Dirac-Frenkel variational principle, which has been used for numerical methods in the field of quantum dynamics for a long time [10, 17, 26, 34, 28] and for dynamic-low rank and related solvers [24, 45, 35, 39, 38, 22]. Instead of globally approximating a PDE solution in time, the Dirac-Frenkel variational principle allows a sequential-in-time training that adapts a nonlinear parameterization, such as a neural network, over time. In contrast to classical numerical methods in vector spaces, the approximate solution in Dirac-Frenkel schemes is allowed to depend nonlinearly on its parameters and so to lie on a smooth manifold. The update to the nonlinear parameterization is calculated at each time step according to the orthogonal projection of the dynamics onto the tangent space of the manifold induced by the nonlinear parameterization. The Dirac-Frenkel variational principle has been adapted for the nonlinear approximation of PDEs with neural networks. In particular [14, 1, 7] formulate a sequential-in-time method based on the Dirac-Frenkel variational principle.

The neural network represents the PDE solution at a point in time. The time-dependence then arises by allowing the parameters—the weights and biases of the network—to vary in time. The network parameters are then evolved forward according to the time dynamics which govern the PDE. This is in contrast to global-in-time methods, in which time enters the network as an additional input variable. By construction, an approximate solution obtained with a sequential-in-time method is causal, in that the solution at future times depends only on the solution at the current time.

Although these methods have demonstrated success in solving various PDEs [34, 14, 7, 54, 16, 8, 30], there are open challenges: First, the sequential-in-time training is prone to overfitting which can lead to a quick accumulation of the residual over time. Second, the local training step has to be repeated at each time step, which can be computationally costly, especially with direct solvers that have costs increase quadratically with the number of network parameters. The work [16] proposes to address the two issues by using iterative solvers, instead of direct ones, and by re-training the network occasionally over the sequential-in-time training. We show with numerical experiments below that the re-training of the network can be computationally expensive. Additionally, the performance of iterative solvers depends on the condition of the problem, which can be poor in the context of sequential-in-time training.

**Our approach and contributions: Randomized sparse updates for schemes based on the Dirac-Frenkel variational principle** We build on the previous work in sequential-in-time methods following a similar set up as [7] based on the Dirac-Frenkel variational principle. Where all previous methods solve local training problems that update every parameter of the network at each time step, we propose a modification such that only randomized sparse subsets of network parameters are updated at each time step:

(a) The randomization avoids overfitting locally in time and so helps preventing the error from accumulating quickly over the sequential-in-time training, which is motivated by dropout that addresses a similar issue of overfitting due to neuron co-adaptation.

(b) The sparsity of the updates reduces the computational costs of training without losing expressiveness because many of the network parameters are redundant locally at each time step.

Our numerical experiments indicate that the proposed scheme is up to two orders of magnitude more accurate at a fixed computational cost and up to two orders of magnitude faster at a fixed accuracy.

We release our code implementation here: `https://github.com/julesberman/RSNG`

## 2 Sequential-in-time training for solving PDEs

### 2.1 Evolution equations, Dirac-Frenkel variational principle, Neural Galerkin schemes

Given a spatial domain $\mathcal{X} \subseteq \mathbb{R}^d$, and a time domain $\mathcal{T} = [0, T) \subseteq \mathbb{R}$, we consider a solution field $u : \mathcal{T} \times \mathcal{X} \to \mathbb{R}$ so that $u(t, \cdot) : \mathcal{X} \to \mathbb{R}$ is in a function space $\mathcal{U}$ at each time $t$, with dynamics

$$
\begin{aligned}
\partial_t u(t, \boldsymbol{x}) &= f(\boldsymbol{x}, u) && \text{for } (t, \boldsymbol{x}) \in \mathcal{T} \times \mathcal{X} \\
u(0, \boldsymbol{x}) &= u_0(\boldsymbol{x}) && \text{for } \boldsymbol{x} \in \mathcal{X}
\end{aligned}
\tag{1}
$$

where $u_0 \in \mathcal{U}$ is the initial condition and $f$ can include partial derivatives of $u$ to represent PDEs. We focus in this work on Dirichlet and periodic boundary conditions but the following approach

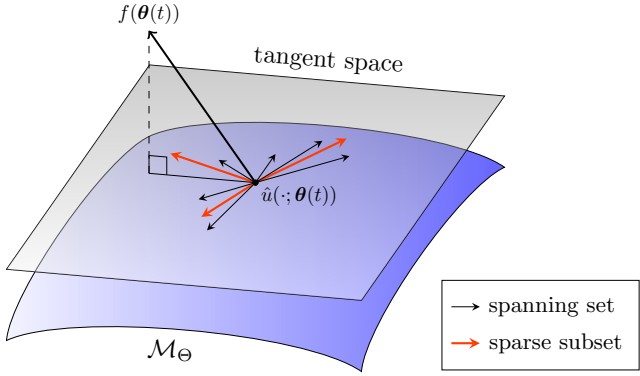

Figure 1: We propose Neural Galerkin schemes that update randomized sparse subsets of network parameters with the Dirac-Frenkel variational principle. Randomization avoids overfitting locally in time, which leads to more accurate approximations than dense updates. Sparsity reduces the computational costs of training without losing expressiveness because many parameters are redundant locally in time.

can be applied with, e.g., Neumann boundary conditions as well [7]. One approach for imposing Dirichlet boundary conditions is by choosing parameterizations that satisfy the boundary conditions by definition [48].

Sequential-in-time training methods approximate $u$ with a nonlinear parameterization such as a neural network $\hat{u} : \mathcal{X} \times \Theta \to \mathbb{R}$, where the parameter vector $\boldsymbol{\theta}(t) \in \Theta \subseteq \mathbb{R}^p$ depends on time $t$; the parameter $\boldsymbol{\theta}(t)$ has $p$ components and enters nonlinear in the second argument of $\hat{u}$. The residual of (1) at time $t$ is

$$r(\boldsymbol{x}; \boldsymbol{\theta}(t), \dot{\boldsymbol{\theta}}(t)) = \nabla_{\boldsymbol{\theta}} \hat{u}(\boldsymbol{x}; \boldsymbol{\theta}(t)) \cdot \dot{\boldsymbol{\theta}}(t) - f\left(\boldsymbol{x}, \hat{u}(\cdot; \boldsymbol{\theta}(t))\right) , \tag{2}$$

where we applied the chain rule to $\partial_t \hat{u}(\cdot; \boldsymbol{\theta}(t))$ to formally obtain $\dot{\boldsymbol{\theta}}(t)$. Methods based on the Dirac-Frenkel variational principle [10, 17, 34] seek $\dot{\boldsymbol{\theta}}(t)$ such that the residual norm is minimized, which leads to the least-squares problem

$$\min_{\dot{\boldsymbol{\theta}}(t)} \|\nabla_{\boldsymbol{\theta}(t)} \hat{u}(\cdot; \boldsymbol{\theta}(t)) \dot{\boldsymbol{\theta}}(t) - f(\cdot; \hat{u}(\cdot; \boldsymbol{\theta}(t)))\|_{L^2(\mathcal{X})}^2 , \tag{3}$$

in the $L^2(\mathcal{X})$ norm $\| \cdot \|_{L^2(\mathcal{X})}$ over $\mathcal{X}$. The least-squares problem (3) gives a $\dot{\boldsymbol{\theta}}(t)$ such that the residual is orthogonal to the tangent space at $\hat{u}(\cdot; \boldsymbol{\theta}(t))$ of the manifold $\mathcal{M}_\Theta = \{\hat{u}(\cdot; \boldsymbol{\theta}) \,|\, \boldsymbol{\theta} \in \Theta\}$ induced by the parameterization $\hat{u}$; see Figure 1. Schemes that solve (3) over time have also been termed Neural Galerkin schemes [7] because (3) can be derived via Galerkin projection as well.

The tangent space at $\hat{u}(\cdot; \boldsymbol{\theta}(t))$ is spanned by the spanning set $\{\partial_{\theta_i} \hat{u}(\cdot; \boldsymbol{\theta}(t))\}_{i=1}^p$, which are the component functions of the gradient $\nabla_{\boldsymbol{\theta}(t)} \hat{u}(\cdot; \boldsymbol{\theta}(t))$; it is important to stress that $\{\partial_{\theta_i} \hat{u}(\cdot; \boldsymbol{\theta}(t))\}_{i=1}^p$ is *not* necessarily a basis of the tangent space because it can contain linearly dependent functions and be non-minimal. The least-squares problem (3) can be realized by assembling a matrix whose columns are the gradient sampled at $n \gg p$ points $\boldsymbol{x}_1, \dots, \boldsymbol{x}_n \in \mathcal{X}$ resulting in $J(\boldsymbol{\theta}(t)) = [\nabla_{\boldsymbol{\theta}(t)} \hat{u}(\boldsymbol{x}_1; \boldsymbol{\theta}(t)), \dots, \nabla_{\boldsymbol{\theta}(t)} \hat{u}(\boldsymbol{x}_n; \boldsymbol{\theta}(t))]^T \in \mathbb{R}^{n \times p}$, which is a batch Jacobian matrix to which we refer to as Jacobian for convenience in the following. Additionally, we form the right-hand side vector $\boldsymbol{f}(\boldsymbol{\theta}(t)) = [f(\boldsymbol{x}_1; \boldsymbol{\theta}(t)), \dots, f(\boldsymbol{x}_n; \boldsymbol{\theta}(t))]^T \in \mathbb{R}^n$ and thus the least-squares problem

$$\min_{\dot{\boldsymbol{\theta}}(t)} \|J(\boldsymbol{\theta}(t)) \dot{\boldsymbol{\theta}}(t) - \boldsymbol{f}(\boldsymbol{\theta}(t))\|_2^2 . \tag{4}$$

The choice of the points $\boldsymbol{x}_1, \dots, \boldsymbol{x}_n$ is critical so that solutions of (4) are good approximations of solutions of (3); however, the topic of selecting the points $\boldsymbol{x}_1, \dots, \boldsymbol{x}_n$ goes beyond this work here and so we just note that methods for selecting the points exist [1, 7] and that we assume in the following that we select sufficient points with $n \gg p$ to ensure that solutions of (4) are good approximations of solutions of (3)

## 2.2 Problem Formulation

**Challenge 1: Parameters are redundant locally in time.** Typically, parameterizations $\hat{u}$ based on deep neural networks lead to Jacobian matrices $J(\boldsymbol{\theta}(t))$ that are low rank in the least-squares problem (4); see, e.g., [37] and Figure 2(a). In our case, a low-rank matrix $J(\boldsymbol{\theta}(t))$ means that components in $\dot{\boldsymbol{\theta}}(t)$ are redundant, because we assume that the samples $\boldsymbol{x}_1, \dots, \boldsymbol{x}_n$ are sufficiently rich. Even if $J(\boldsymbol{\theta}(t))$ is low rank and thus the components in $\dot{\boldsymbol{\theta}}(t)$ are redundant, the problem (4) can still be

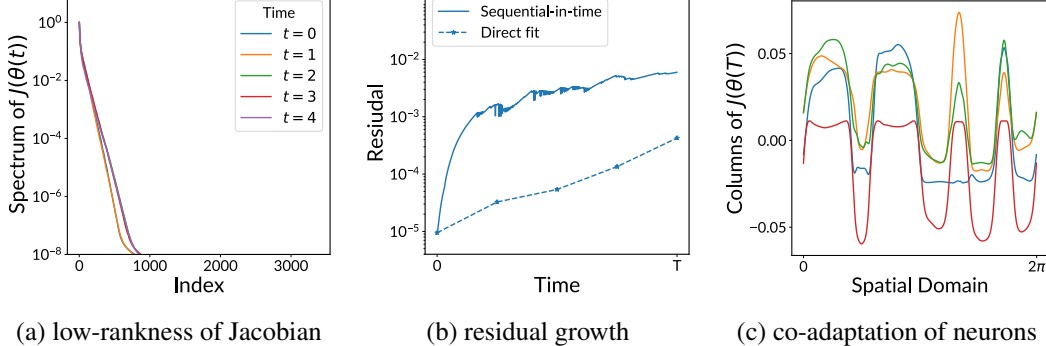

(a) low-rankness of Jacobian       (b) residual growth       (c) co-adaptation of neurons

Figure 2: (a) Jacobians that have low rank locally in time imply that there are redundant parameters in the neural network, which motivates the proposed sparse updates that lead to speedups without losing expressiveness. (b) The residual grows quickly with sequential-in-time training (and dense updates). This is not due to a limitation with the expressiveness of the network because directly fitting the network to solutions indicates that there exist other parameters that can lead to lower residuals. (c) Sequential-in-time training (with dense updates) results in co-adapted neurons as indicated by the highly correlated columns of the $J$ matrix. Plots for experiment with Allen-Cahn equation (Sec. 4).

solved with standard linear algebra methods such as the singular value decomposition (SVD) because they compress the matrix $J(\boldsymbol{\theta}(t))$ and regularize for, e.g., the minimal-norm solution; however, the costs of performing the SVD to solve (4) scales as $\mathcal{O}(np^2)$, and thus is quadratic in the number of parameters $p$. This means that a redundancy in $\dot{\boldsymbol{\theta}}(t)$ of a factor two leads to a $4\times$ increase in the computational costs. Note that the problem typically is poorly conditioned because $J(\boldsymbol{\theta}(t))$ is low rank, which makes the direct application of iterative solvers challenging.

**Challenge 2: Overfitting leads to high residual over time.** The residual from solving (4) can rapidly increase over time which in turn increases the overall error. This indicates that the tangent space along the trajectory $\boldsymbol{\theta}(t)$ becomes ill suited for approximating the right-hand side vector $\boldsymbol{f}(\boldsymbol{\theta}(t))$ in (4). We compare the residual of the least-squares problem (4) that is obtained along a trajectory of $\boldsymbol{\theta}(t)$ from sequential-in-time training with the schemes above to the residual of (4) from a network that is fit to the true solution at each point in time; details in Appendix A.1. As shown in Figure 2(b), a lower residual is achieved by the network that is fit to the true solution.

We aim to understand this phenomenon through the lens of overfitting: the sequential-in-time training can be thought of as successive fine-tuning, in the sense that at each time step we must make a small update to our parameters to match the solution at the next time step. However, fine-tuning is well known to be prone to over-fitting and model degeneration [3]. In the setting considered in this work, overfitting means that the representation $\hat{u}(\cdot; \boldsymbol{\theta}(t))$ does not generalize well to the next time step. Not generalizing well means that a local change to $\boldsymbol{\theta}(t)$ is insufficient to move $\hat{u}(\cdot; \boldsymbol{\theta}(t))$ according to the desired update given by $\dot{\boldsymbol{\theta}}(t)$ to match the right-hand side $\boldsymbol{f}(\boldsymbol{\theta}(t))$, which implies that a large residual is incurred when solving (4). A common approach to prevent overfitting is dropout [47], especially when applied to fine-tuning tasks with dropout variants proposed in [3, 29], while other approaches are formulated specifically around sparse updates [49, 52]. Dropout is motivated by the observation that dense updates to parameters in neural networks can cause overfitting by leading neurons to co-adapt. Typically, co-adaptation is characterized by layer-wise outputs with high covariance [9]. In the case of sequential-in-time training with the schemes discussed above, co-adaptation implies the columns of the Jacobian matrix $J(\boldsymbol{\theta}(t))$ are correlated and thus close to linearly dependent. So as neurons co-adapt, component functions of the gradient become redundant and may be less suited for approximating $\boldsymbol{f}(\boldsymbol{\theta}(t))$ causing the high residual for the least-squares problem; see Figure 2(b). This could also be characterized by the ill conditioning issue pointed out in [16]. We see empirical evidence of co-adaptation in Figure 2(c), where we plot component functions of the gradient and see that they are strongly correlated at the end time $T$.

# 3 Randomized Sparse Neural Galerkin (RSNG) schemes

We introduce randomized sparse Neural Galerkin (RSNG) schemes that build on the Dirac-Frenkel variational principle to evolve network parameters $\boldsymbol{\theta}(t)$ sequentially over time $t$ but update only sparse subsets of the components of $\boldsymbol{\theta}(t)$ and randomize which components of $\boldsymbol{\theta}(t)$ are updated. The sparse updates reduce the computational costs of solving the least-squares problem (4) while taking advantage of the low rank structure of $J(\boldsymbol{\theta})$ which indicates components of the time derivative $\dot{\boldsymbol{\theta}}(t)$ are redundant and can be ignored for updating $\boldsymbol{\theta}(t)$ without losing expressiveness. The randomization of which components of $\boldsymbol{\theta}(t)$ are updated prevents the overfitting described above.

## 3.1 Randomized sketch of residual

To define the sketch matrix $S_t$, let $\boldsymbol{e}_1, \ldots, \boldsymbol{e}_p$ be the $p$-dimensional canonical unit vectors so that $\boldsymbol{e}_i$ has entry one at component $i$ and zeros at all other components. We then define $s$ independent and identically distributed random variables $\xi_1(t), \ldots, \xi_s(t)$ that depend on time $t$. The distribution of $\xi_i(t)$ is $\pi$, which is supported over the set of indices $\{1, \ldots, p\}$. The random matrix $S_t$ of size $p \times s$ is then defined as $S_t = [\boldsymbol{e}_{\xi_1(t)}, \ldots, \boldsymbol{e}_{\xi_s(t)}]$. The corresponding sketched residual analogous to (2) is

$$r_s(\boldsymbol{x}; \boldsymbol{\theta}(t), \dot{\boldsymbol{\theta}}_s(t)) = \nabla_{\boldsymbol{\theta}} \hat{u}(\boldsymbol{x}; \boldsymbol{\theta}(t)) S_t \dot{\boldsymbol{\theta}}_s(t) - f(\boldsymbol{x}, \hat{u}(\cdot; \boldsymbol{\theta}(t))), \tag{5}$$

where now $\dot{\boldsymbol{\theta}}_s(t) \in \mathbb{R}^s$ is of dimension $s \ll p$.

## 3.2 Projections onto randomized approximations of tangent spaces

Using the sketch matrix $S_t$, we obtain from the spanning set $\{\partial_{\theta_i} \hat{u}(\cdot; \boldsymbol{\theta}(t))\}_{i=1}^p$ of component functions of $\nabla_{\boldsymbol{\theta}} \hat{u}(\cdot; \boldsymbol{\theta}(t))$ a subset $\{\partial_{\theta_{\xi_i(t)}} \hat{u}(\cdot; \boldsymbol{\theta}(t))\}_{i=1}^s$ with $s$ functions. The set $\{\partial_{\theta_{\xi_i(t)}} \hat{u}(\cdot; \boldsymbol{\theta}(t))\}_{i=1}^s$ spans at least approximately the tangent space at $\hat{u}(\cdot; \boldsymbol{\theta}(t))$ of $\mathcal{M}_\Theta$ but has only $s \ll p$ elements. The motivation is that the full spanning set $\{\partial_{\theta_i} \hat{u}(\cdot; \boldsymbol{\theta}(t))\}_{i=1}^p$ contains many functions that are close to linearly dependent (Jacobian is low rank) and thus sub-sampling the component functions still gives reasonable tangent space approximations that preserves much of the expressiveness; see Figure 1. While the low rankness depends on the complexity of the problem and parametrization, we observe low rankness in all our examples; see Appendix A.1 for further discussion.

We now introduce a least-squares problem based on the sparse spanning set $\{\partial_{\theta_{\xi_i(t)}} \hat{u}(\cdot; \boldsymbol{\theta}(t))\}_{i=1}^s$ that is analogous to the least-squares problem problem based on the full spanning set given in (4). We seek $\dot{\boldsymbol{\theta}}_s(t) \in \mathbb{R}^s$ with $s$ components that solves

$$\min_{\dot{\boldsymbol{\theta}}_s(t) \in \mathbb{R}^s} \|\nabla_{\boldsymbol{\theta}} \hat{u}(\cdot; \boldsymbol{\theta}(t)) S_t \dot{\boldsymbol{\theta}}_s(t) - f(\cdot; \hat{u}(\cdot; \boldsymbol{\theta}(t)))\|_{L^2(\mathcal{X})}^2. \tag{6}$$

To obtain $\dot{\boldsymbol{\theta}}(t)$ to update $\boldsymbol{\theta}(t)$, we set $\dot{\boldsymbol{\theta}}(t) = S_t \dot{\boldsymbol{\theta}}_s(t)$. Thus, the components of $\dot{\boldsymbol{\theta}}(t)$ that are selected by $S_t$ are set to the corresponding value of the component of $\dot{\boldsymbol{\theta}}_s(t)$ and all other components are set to zero, which means that the corresponding components of $\boldsymbol{\theta}(t)$ are not updated. We can realize (6) the same way as the full least-squares problem in (4) by using the full Jacobian matrix and $S_t$ to define the sparse Jacobian matrix as $J_s(\boldsymbol{\theta}(t)) = J(\boldsymbol{\theta}(t)) S_t$ and the right-hand side vector $\boldsymbol{f}(\boldsymbol{\theta}(t))$ analogous to Section 2 to obtain the discrete least-squares problem

$$\min_{\dot{\boldsymbol{\theta}}_s(t)} \|J_s(\boldsymbol{\theta}(t)) \dot{\boldsymbol{\theta}}_s(t) - \boldsymbol{f}(\boldsymbol{\theta}(t))\|_2^2. \tag{7}$$

The choice of the distribution $\pi$ is critical and depends on properties of the Jacobian matrix $J(\boldsymbol{\theta}(t))$. Distributions based on leverage scores provide tight bounds with regard to the number of columns one needs to sample in order for the submatrix to be close to an optimal low rank approximation of the full matrix with high probability [13]. But these distributions can be expensive to sample from. Instead, uniform sampling provides a fast alternative.

The number of columns will not grow too quickly if the full matrix is sufficiently incoherent [19]. This means some columns do not carry a disproportionate amount of information relative to other columns. We numerically see that in our case the Jacobian matrix $J(\boldsymbol{\theta}(t))$ is sufficiently incoherent. Thus we can choose a uniform distribution over the set of indices $\{1, \ldots, p\}$ to get the benefits of low rank approximation in a computationally efficient way.

---

**Algorithm 1** Randomized Neural Galerkin scheme with sparse updates

---

Fit parameterization $\hat{u}(\cdot; \boldsymbol{\theta}^{(0)})$ to initial condition $u_0$ to obtain $\boldsymbol{\theta}^{(0)}$
**for** $k = 1, \ldots, K$ **do**
    Draw realization of sketching matrix $S_k$ as described in Section 3.1
    Solve for sparse update $\Delta\boldsymbol{\theta}_s^{(k-1)}$ with least-squares problem (9)
    Lift sparse update $\Delta\boldsymbol{\theta}^{(k-1)} = S_k \Delta\boldsymbol{\theta}_s^{(k-1)}$
    Update $\boldsymbol{\theta}^{(k)} = \boldsymbol{\theta}^{(k-1)} + \delta t \Delta\boldsymbol{\theta}^{(k-1)}$
**end for**

---

### 3.3  Discretizing in time

We discretize the time interval $\mathcal{T}$ with $K \in \mathbb{N}$ regularly spaced time steps $0 = t_0 < t_1 < \cdots < t_K = T$ with $\delta t = t_k - t_{k-1}$ for $k = 1, \ldots, K$. At time $t_0$, we obtain $\boldsymbol{\theta}^{(0)} \in \mathbb{R}^p$ by fitting the initial condition $u_0$. We then update

$$\boldsymbol{\theta}^{(k)} = \boldsymbol{\theta}^{(k-1)} + \delta t \Delta\boldsymbol{\theta}^{(k-1)} \tag{8}$$

for $k = 1, \ldots, K$ so that $\boldsymbol{\theta}^{(k)}$ is the time-discrete approximation of $\boldsymbol{\theta}(t_k)$ and thus $\hat{u}(\cdot; \boldsymbol{\theta}^{(k)})$ approximates the solution $u$ at time $t_k$. The sparse update $\Delta\boldsymbol{\theta}_s^{(k-1)}$ approximates $\dot{\boldsymbol{\theta}}_s(t_{k-1})$ and is obtain by the time-discrete counterpart of (7), which is given by

$$\min_{\Delta\boldsymbol{\theta}_s^{(k-1)}} \|J_s(\boldsymbol{\theta}^{(k-1)}) S_k \Delta\boldsymbol{\theta}_s^{(k-1)} - \boldsymbol{f}(\boldsymbol{\theta}^{(k-1)})\|_2^2, \tag{9}$$

if time is discretized with the forward Euler method. Other discretization schemes can be used as well, which then lead to technically more involved problems (9) that remain conceptually similar though. The sparse update is lifted to $\Delta\boldsymbol{\theta}^{(k-1)} = S_k \Delta\boldsymbol{\theta}_s^{(k-1)}$ so that the update (8) can be computed.

### 3.4  Computational procedure of RSNG

We describe the proposed RSNG procedure in algorithmic form in Algorithm 1. We iterate over the time steps $k = 1, \ldots, K$. At each time step, we first sketch the Jacobian matrix by creating a submatrix from randomly sampled columns. Notice that $S_k$ need not actually be assembled as its action on the Jacobian matrix can be accomplished by indexing. We then solve the least-squares problem given in (9) using our sketched Jacobian to obtain $\Delta\boldsymbol{\theta}_s^{(k-1)}$. A direct solve of this system dominates the computational cost of making a time step and scales in $O(ns^2)$ time. The components of $\Delta\boldsymbol{\theta}^{(k-1)}$ corresponding to the indices that have not been selected are filled with zeros. We then update the parameter $\boldsymbol{\theta}^{(k-1)}$ to $\boldsymbol{\theta}^{(k)}$ via $\Delta\boldsymbol{\theta}^{(k-1)}$.

The whole integration process scales as $O(\frac{T}{\delta t} ns^2)$ in time.

## 4  Numerical experiments

We demonstrate RSNG on a wide range of evolution equations, where speedups of up to two orders of magnitude are achieved compared to comparable schemes with dense updates. We also compare to global-in-time methods, where we achieve up to two orders of magnitude higher accuracy.

### 4.1  Setup and equations

**Examples**    We now describe the details of the PDEs that we use to evaluate our method. We choose these particular setups to test our method on a diverse set of challenges including problems with global and local dynamics and solutions with sharp gradients and fine grained details. For visualization of the solutions of these equations see the Appendix A.4.

*Reaction-diffusion problem modeled by Allen-Cahn (AC) equation:* The Allen-Cahn equation models prototypical reaction diffusion phenomena and is given as,

$$\partial_t u(t, x) = \epsilon \partial_{xx} u(t, x) + u(t, x) - u(t, x)^3.$$

We choose $\epsilon = 5 \times 10^{-3}$, with periodic boundary condition $\mathcal{X} = [0, 2\pi)$ and initial condition,

$$u_0(x) = \frac{1}{3}\tanh(2\sin(x)) - \exp(-23.5(x - \frac{\pi}{2})^2) + \exp(-27(x - 4.2)^2) + \exp(-38(x - 5.4)^2)).$$

This initial condition results in challenging dynamics that are global over the spatial domain.

*Flows with sharp gradients described by Burgers' equation:* The Burgers' equation is given by,

$$\partial_t u(t, x) = \epsilon \partial_{xx} u(t, x) - u(t, x)\partial_x u(t, x).$$

We choose $\epsilon = 1 \times 10^{-3}$, with periodic boundary condition $\mathcal{X} = [-1, 1)$ and initial condition,

$$u_0(x) = (1 - x^2)\exp(-30(x + 0.5)^2).$$

The corresponding solution field has sharp gradients that move in the spatial domain over time, which can be challenging to approximate.

*Charged particles in electric field:* The Vlasov equation describes the time evolution of collisionless charged particles under the influence of an electric field. The equation models the distribution of such particles in terms of their position and velocity. We consider the case of one position dimension and one velocity dimension, making our domain $\mathcal{X} \subseteq \mathbb{R}^2$. The equation is given by,

$$\partial_t u(t, x, v) = -v\partial_x u(t, x, v) + \partial_x \phi(x)\partial_v u(t, x, v)$$

where $x$ is the position, $v$ is the velocity and $\phi$ is the electric field. We consider the case with periodic boundary condition $\mathcal{X} = [0, 2\pi) \times [-6, 6)$ and initial condition,

$$u_0(x, v) = \frac{1}{\sqrt{2\pi}}\exp(\frac{-v^2}{2})$$

with a fixed electric field $\phi(x) = \cos(x)$. This particular setup evolves into a distribution with fine grained details along a separatrix surrounding the potential well.

**Setup** We parameterize with a feed-forward multi-layer perceptron. All our networks use linear layers of width 25 followed by non-linear activation functions, except the last layer which has no activation and is of width 1 so that $\hat{u}(\boldsymbol{x}, \boldsymbol{\theta}(t)) \in \mathbb{R}$. To vary the number of total parameters $p$, we vary the depth of networks ranging from 3–7 layers. We use rational activation functions which in our experiments allowed for fitting initial conditions faster and more accurately than a standard choice such as $\tanh$ or ReLU [5]. To enforce periodic boundary conditions, we modify the first layer so that it outputs periodic embeddings as in [7]; for details see Appendix A.5. The periodic embedding ensures that the boundary conditions are enforced exactly. For additional details on enforcing other types of boundary conditions (e.g. Dirichlet and Neumann) exactly in neural networks see [12, 7, 48]. We sample points from the domain on an equidistant grid. For time integration we use a RK4 scheme with a fixed time step size. The time step sizes are $5e{-}3$, $1e{-}3$, $5e{-}3$ and we integrate up to end time 4, 4, and 3 for the Allen-Cahn, Burgers', and Vlasov equations, respectively. All error bars show $+/-$ two standard errors over three random realizations which results in different sketching matrices at each time step. Relative errors are computed over the full space-time domain, unless the plot is explicitly over time.

All gradients and spatial derivatives are computed with automatic differentiation implemented in JAX [6]. All computations are done in single precision arithmetic which is the default in JAX. All runtime statistics were computed on the same hardware, a Nvidia Tesla V100 w/ 32 GB memory. All additional hyperparameters are described in Appendix A.5.

### 4.2 Results

**RSNG achieves higher accuracy than schemes with dense updates at same computational costs** In Figure 3 we plot the relative error over time. The curves corresponding to "dense updates" use a 3 layer network and integration is performed using dense updates. For RSNG, we use a 7 layer network and integrate with sparse updates, setting the number of parameters we update, $s$, equal to the total number of parameters in the 3 layer network and thus equal to the number of parameters that are updated by "dense updates." Thus, the comparison is at a fixed computational cost. The error achieved with RSNG is one to two orders of magnitude below the error obtained with dense updates, across all examples that we consider. In Figure 4(a), we see that as we increase the network size,

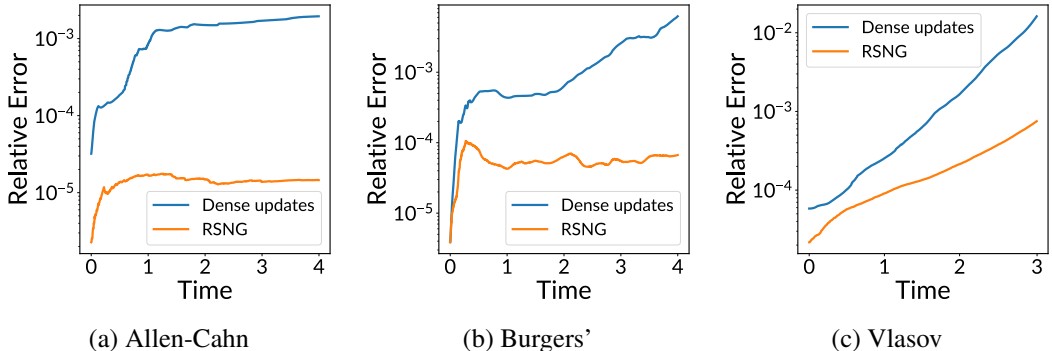

|                  |                  |                  |
| :--------------: | :--------------: | :--------------: |
| (a) Allen-Cahn   | (b) Burgers'     | (c) Vlasov       |

Figure 3: We plot the relative error over time for RSNG versus dense updates at $s = 757$. We see RSNG leads to orders of magnitude lower errors than dense updates for the same costs.

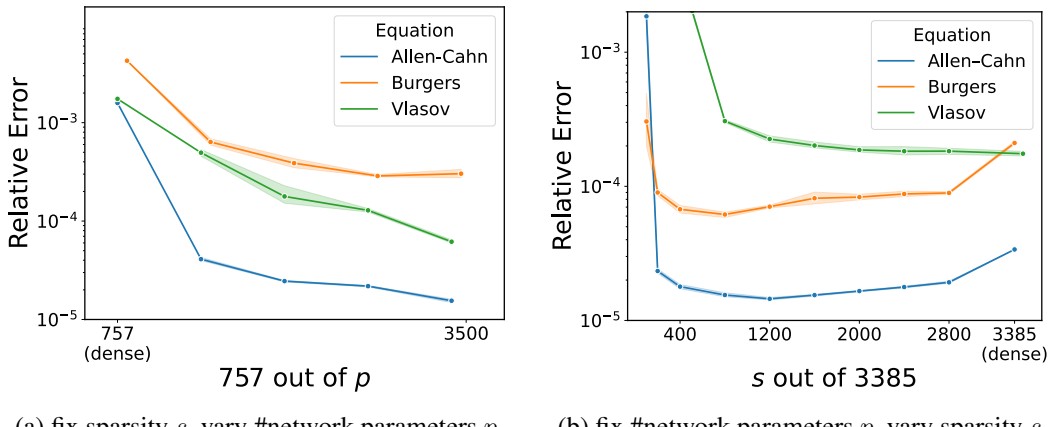

(a) fix sparsity $s$, vary #network parameters $p$      (b) fix #network parameters $p$, vary sparsity $s$

Figure 4: (a) RSNG benefits from the additional expressiveness of larger networks (larger $p$) while only using a fixed number of parameters (fixed $s$) at each time step. (b) As we decrease the number of parameters $s$ in the sparse update, but keep the total number of parameters $p$ of the network the same, we achieve lower errors than dense updates. Thus, RSNG outperforms dense updates while incurring lower computational costs. Error bars generated over random sketch matrices, $S_t$.

the relative error decreases as the sparse updates allow us to exploit the greater expressiveness of larger networks while incurring no additional computational cost in computing (9). But we note that increasing the size of the full network will make computations of $J(\boldsymbol{\theta})$ and $\boldsymbol{f}(\boldsymbol{\theta})$ more expensive because of higher costs of computing gradients. However, for the network sizes that we consider in this work, this effect is negligible compared to the cost of solving (9).

**RSNG achieves speedups of up to two orders of magnitude** In Figure 5(a), we compare the runtime of RSNG to the runtime of a scheme with dense updates that uses a direct solver and to the runtime of a scheme with dense updates that uses an iterative solver as proposed in [16]. The time is computed for Burgers' equation and the sparsity $s$ of RSNG is chosen such that all methods reach a comparable level of error. We find that RSNG is faster than direct solves with dense updates by two orders of magnitude and faster than the iterative solver by one order of magnitude.

The results show that while using an iterative solver as in [16] does speed up the method relative to direct solves with dense updates, it can still be quite slow for networks with many parameters $p$. Additionally, convergence of the iterative method given in [16] requires a number of hyperparameters to be chosen correctly, which may require an expensive search or a priori knowledge about the solution. Note that our RSNG method does not preclude the use of an iterative method to speed up the least-squares solves further.

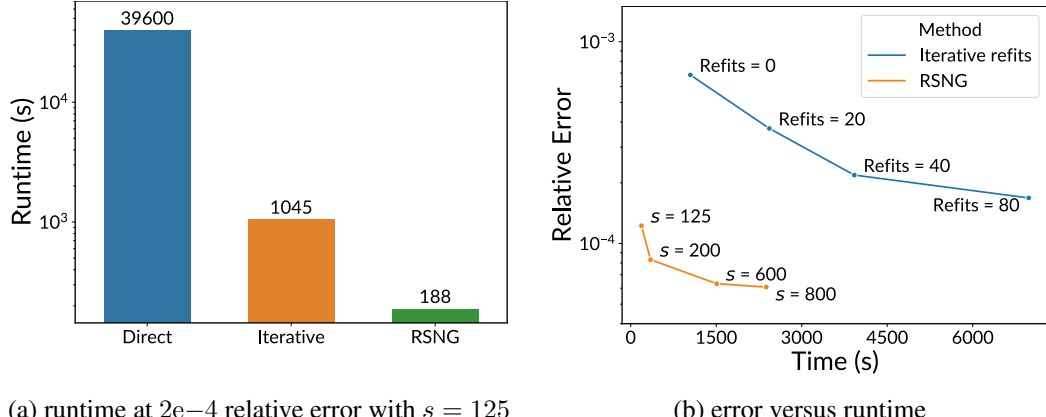

(a) runtime at $2\mathrm{e}-4$ relative error with $s = 125$

(b) error versus runtime

Figure 5: RSNG has lower computational costs than dense updates with direct and iterative least-squares solvers. Plots for numerical experiment with Burgers' equation

To address the overfitting problem, the work [16] refits the network to the current approximate solution from a random initialization periodically during time integration. In Figure 5(b), we show the relative error versus the runtime for the iterative solver with various numbers of refits and for RSNG at different sparsity $s$. While refitting the network can reduce the relative error, it incurs a high computational cost. By contrast, for appropriate sparsity $s$, RSNG outperforms the method given in [16] in both speed and accuracy.

**Varying sparsity $s$ at fixed number of total parameters $p$ in network**  We now study the effect of varying the sparsity $s$ (i.e., number of parameters updated by sparse updates) for a fixed network of total size $p$, in this case a 7 layer network. In Figure 4(b), we see that for a network of fixed size, sparse updates can reduce the relative error by about 2–3× when compared to dense updates. This is notable as the computational costs decrease quadratically with $s$. Thus, the combination of sparsity and randomized updates in RSNG can deliver both improved performance and lower computational cost. We see that at the beginning, when the number $s$ is too small, the expressiveness suffers and the error becomes large. This is because if $s$ is less than the rank of the dense Jacobian then the sparsified Jacobian will necessarily have less representational power. However, we stress that RSNG is robust with respect to $s$ in the sense that for a wide range of $s$ values the error is lower than for dense updates.

The high error when performing dense updates $s = p$ in Figure 4(b) for Allen-Cahn and Burgers' equation is due to the overfitting problem described in Section 2.2. As updates become denser, the method is more likely to overfit to regions of the parameter space in which the Jacobian, $J(\boldsymbol{\theta})$, is ill suited for approximating the right-hand side $f$ at future time steps (see Section 2). We can see this explicitly in Figure 6 where we plot the residual over time for sparse and dense updates on the Allen-Cahn equation. Initially, the dense updates lead to a lower residual. This makes sense as they begin at the same region of parameters space. But as the two methods navigate to different regions of parameters space, we see RSNG begins to incur a lower residual relative to dense updates. This indicates that RSNG ameliorates the problem of overfitting and so leads to a lower residual as shown in Figure 6(b).

**Comparison with global-in-time methods**  We compare our method to global-in-time methods which aim to globally minimize the PDE residual over the entire space-time domain. We compare to the original PINN formulation given in [41]. Additionally we compare to a variant termed Causal PINNs, which impose a weak form of time dependence through the loss function [51]. We select this variant as it claims to have state of the art performance among PINNs on problems such as the Allen-Cahn equation. In Table 1, we see that our sequential-in-time RSNG method achieves a higher accuracy by at least one order of magnitude compared to PINNs. Additionally, in terms of computational costs, RSNG outperforms both PINN variants, as their global-in-time training is expensive and requires many residual evaluations. We note that the training time of PINNs is directly

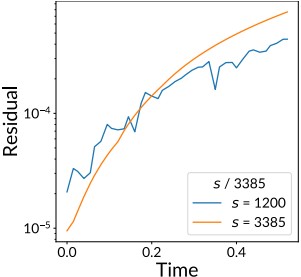

(a) residual at early times

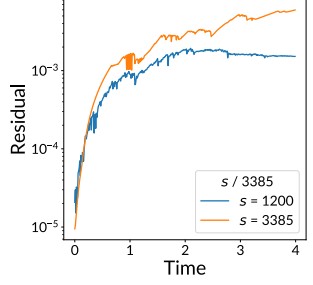

(b) residual over full time

Figure 6: Plot (a) shows the residual of dense and sparse updates at early time steps. Initially, dense updates must have a lower residual as $JS_t$ spans a subspace of the tangent space given by $J$. But in plot (b), we see that after a few time steps, dense updates overfit and the residual grows quicker than with sparse updates.

| PDE | Method | $L^2$ Relative Error | Time(s) | $s$ |
|---|---|---|---|---|
| Allen-Cahn | PINN | 6.85e−2 | 841 | N/A |
| Allen-Cahn | Causal PINN | 3.84e−4 | 3060 | N/A |
| Allen-Cahn | RSNG | **1.66e−5** | 776 | 800 |
| Allen-Cahn | RSNG | 5.34e−5 | **63** | 150 |
| Burgers' | PINN | 2.34e−3 | 3451 | N/A |
| Burgers' | Causal PINN | 5.19e−4 | 23027 | N/A |
| Burgers' | RSNG | **6.07e−5** | 2378 | 800 |
| Burgers' | RSNG | 2.05e−4 | **188** | 125 |

Table 1: The sequential-in-time training with RSNG achieves about one order of magnitude higher accuracy than global-in-time methods in our examples. Details on training in Appendix A.6.

dependent on the number of optimization iterations and thus they can be trained faster if one is willing to tolerate even higher relative errors.

## 5 Conclusions, limitations, and future work

In this work, we introduced RSNG that updates randomized sparse subsets of network parameters in sequential-in-time training with the Dirac-Frenkel variational principle to reduce computational costs while maintaining expressiveness. The randomized sparse updates are motivated by a redundancy of the parameters and by the problem of overfitting. The randomized sparse updates have a low barrier of implementation in existing sequential-in-time solvers. The proposed RSNG achieves speedups of up to two orders of magnitude compared to dense updates for computing an approximate PDE solution with the same accuracy.

Current limitations leave several avenues for future research: first, as discussed in 3.2, uniform sampling is only appropriate when the Jacobian matrix is of low coherence. Future work may investigate more sophisticated sampling methods such as leverage score and pivoting elements of rank revealing QR. Second, there are problems for which overfitting with dense updates is less of an issue; e.g., the charged particles example in our work. Note that due to the sparsity of the updates, RSNG still achieves a speedup compared to dense updates for the same accuracy for this example though. However, more work is needed to better understand and mathematically characterize which properties of the problems influence the overfitting issue.

We make a general comment about using neural networks for numerically solving PDEs: The equations discussed in this paper are standard benchmark examples used in the machine-learning literature; however, for these equations, carefully designed classical methods can succeed and often have lower runtimes than methods based on nonlinear parameterizations [20, 36]. While these equations provide an important testing ground to demonstrate methodological improvements, future work will extend these results to domains where classical linear methods struggle, e.g., high-dimensional problems and problems with slowly decaying Kolmogorov $n$-widths [15, 7, 40].

We do not expect that this work has negative societal impacts.

**Acknowledgements**   The authors were partially supported by the National Science Foundation under Grant No. 2046521 and the Office of Naval Research under award N00014-22-1-2728. This work was also supported in part through the NYU IT High Performance Computing resources, services, and staff expertise.

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

# A Appendix

## A.1 Details on Figure 2

For Figure 2a–b we look at quantities generated by fitting a network to the true solution at a point in time $t$. This is done in the same way we fit initial conditions described in A.5, but in this context the target function is taken to be the true solution at time $t$. For Figure 2(a), we compute the Jacobian of the network fitted to the true solution at each point in time and then plot its spectrum. For Figure 2(b), we take the network fitted to the true solution and compute the residual from the least-squares problem 4 to give the data points in the "Direct fit" line.

In Figure 7 we provide versions of Figure 2(a) from the main text but for all the equations considered. Vlasov is a more complex problem that exhibits a less sharp decay but is still distinctly rank deficient.

## A.2 Details on Speed-Up

Here we provide additional results on the speed-up provided by RSNG for different equations. In Figure 8 we provide versions of Figure 5(a) from the main text but for all the equations considered. In Figure 9 we show how the runtime of Neural Galerkin schemes scales with $s$, thus showing the quadratic speed-up provided by RSNG as we reduce $s$.

## A.3 Applications to High Dimensional Problems

Neural Galerkin schemes have been shown to be a useful approach to high dimensional PDEs [7]. In Figure 10 we demonstrate that RSNG may be applicable in these settings we well. We fit a neural network to a numerical solution of the Fokker-Planck equation in 5 dimensions; see [7, Section 5.4.1] for a description of the setup. The results show that for a sufficiently large network, the Jacobian has a low-rank structure as in the examples in the paper and a steep decay in the singular values so that random sketching strategies will likely be successful.

## A.4 Ground Truth

The ground truth for the Allen-Cahn and Burgers' equations were generated using a spectral method with a fourth order integrator implemented in the `spin` solver as part of the Chebfun package in Matlab. We used a spatial grid of $10\,000$ points and time step size of $1e{-}3$.

The ground truth for the Vlasov equation was generated with a 4th order finite difference scheme in space and an RK4 time integration scheme. We sample $10^6$ points over the full 2D space domain and a time step size of $1e{-}3$ was used for time integration.

In Figure 11,12,13 we show plots of the ground truth solutions at the beginning, middle, and end of integration time for the equations we examine. We can see that they display the characteristics described in Section 4.1.

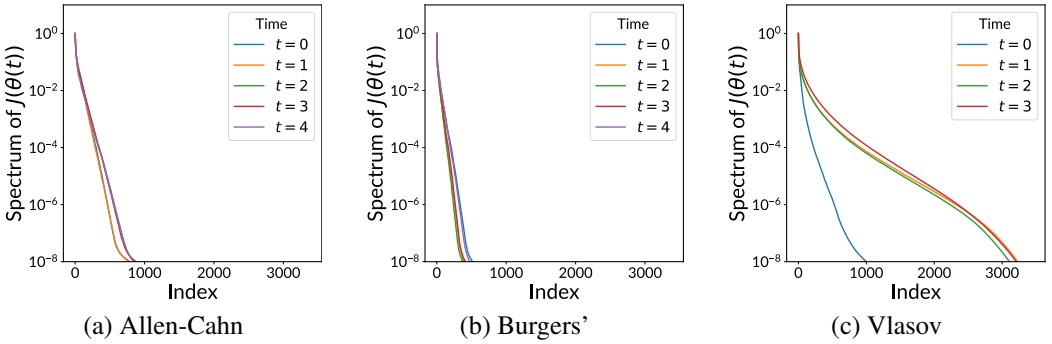

Figure 7: Decay of singular values of $J(\boldsymbol{\theta}(t))$

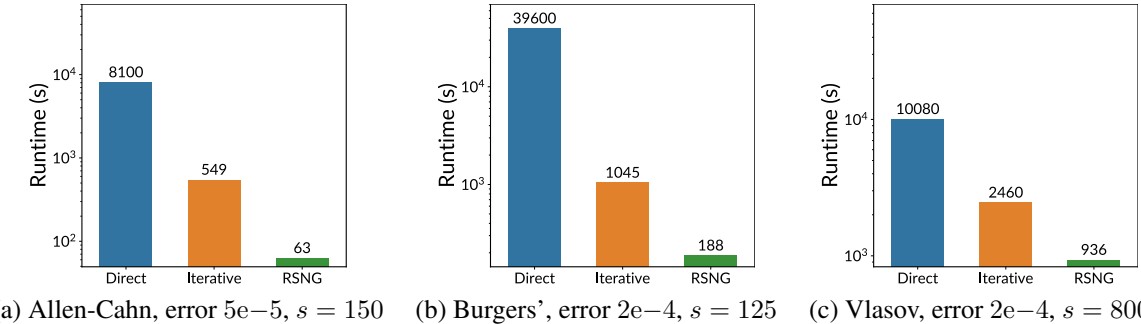

(a) Allen-Cahn, error $5\mathrm{e}{-}5$, $s = 150$    (b) Burgers', error $2\mathrm{e}{-}4$, $s = 125$    (c) Vlasov, error $2\mathrm{e}{-}4$, $s = 800$

Figure 8: Speedups of RSNG over dense updates with direct and iterative solver.

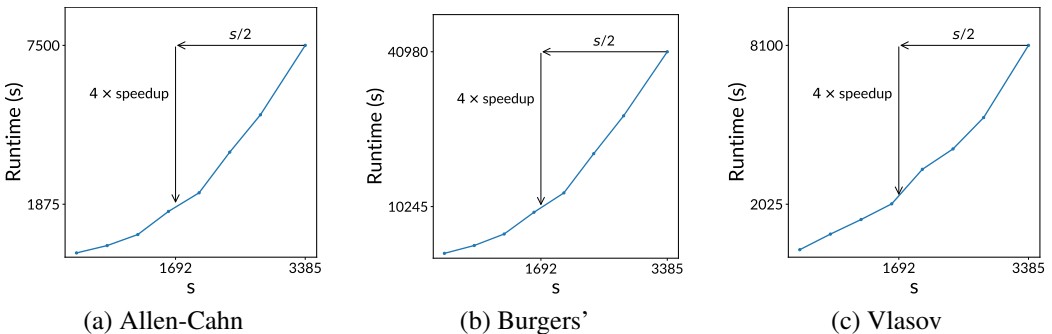

(a) Allen-Cahn    (b) Burgers'    (c) Vlasov

Figure 9: Speedups of RSNG scale quadratic with sparsity $s$.

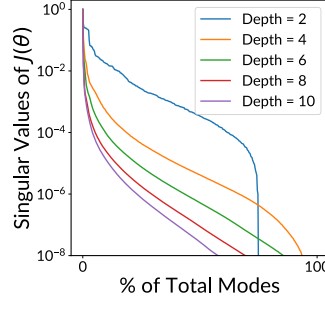

Figure 10: Singular values of the Jacobian of the network fit to a Fokker-Planck solution in five dimension decays quickly too; providing indication that our RSNG approach is applicable in these settings as well.

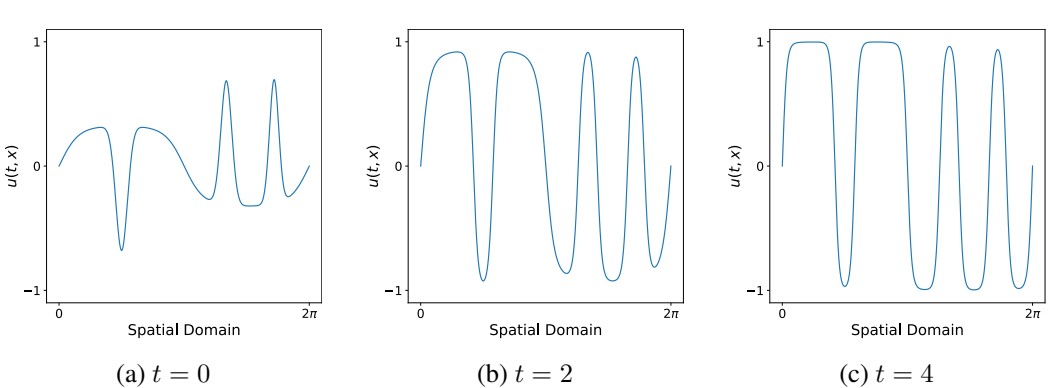

(a) $t = 0$    (b) $t = 2$    (c) $t = 4$

Figure 11: True solution $u(t, x)$ for Allen-Cahn

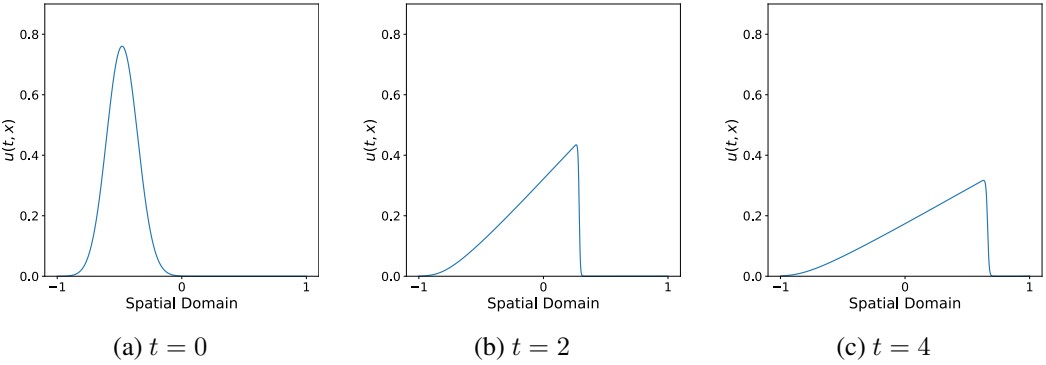

(a) $t = 0$  (b) $t = 2$  (c) $t = 4$

Figure 12: True solution $u(t, x)$ for Burgers'

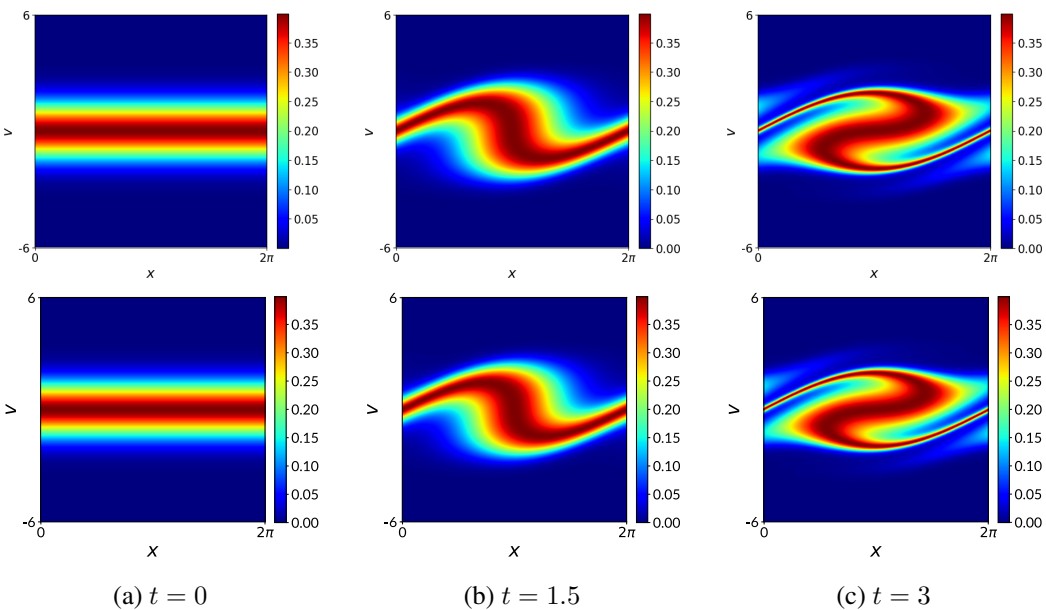

(a) $t = 0$  (b) $t = 1.5$  (c) $t = 3$

Figure 13: True solution (top) vs RSNG solution (bottom) for Vlasov

## A.5 Architecture and Hyperparameters

For an input $x \in \mathbb{R}^d$ a periodic embedding layer with period $P$ is defined with the optimization parameters $a, \phi, b \in \mathbb{R}^d$ as,

$$\texttt{PeriodicEmbed}(x) = \sum_{i=1}^{d} \left[ a \cos(x\frac{2\pi}{P} + \phi) + b \right]_i .$$

This operation is the repeated $w$ times for different parameters $a, \phi, b$ where $w$ denotes the width of the layer resulting in a output vector $y \in R^w$

To fit the initial condition, we minimize the $L^2$ distance between the network and the initial condition as well as the $L^2$ distance between the first derivative of the network and the first derivative of the initial condition with respect to the spatial domain $\mathcal{X}$. We evaluate the loss function over $10\,000$ equispaced points for Allen-Cahn and Burgers' equation and $200\,000$ points for Vlasov. We fit our initial conditions with two nonlinear solvers. First we run `L-BFGS` with $1000$ iterations, then we use an Adam optimizer with the following hyperparameters,

- iterations : $100\,000$
- learning rate : $1e-3$
- scheduler : cosine decay

- decay steps : 500

The number of iterations and the number of points we sample are chosen to fit the initial condition to high accuracy to avoid polluting the results in our analysis with errors of fitting the initial condition.

To assemble the Jacobian matrix $J(\boldsymbol{\theta}(t))$, the gradient of $\hat{u}$ is evaluated on samples generated from the spatial domain. If not noted otherwise, we use $10\,000$ equidistant points for Allen-Cahn and Burgers' and $200\,000$ equidistant points for Vlasov. For the time-optimized RSNG results in Table 1 (rows 4 and 8) we use $1000$ equidistant points for Allen-Cahn and Burgers'. In the dense and sparse least-squares system we regularize the direct solver so as to avoid numerical instability. For this we set the `rcond` parameters in `numpy` implementation of `lstsq`. The values used are $1e{-}4$, $1e{-}4$, $1e{-}5$ for Allen-Cahn, Burgers', and Vlasov respectively.

## A.6  Global Methods Benchmark

Here we detail the training setups for our benchmarks of the global methods given in Table 1.

For all PINN experiments we sampled data on a grid with 100 points in the time domain and 256 points in the spatial domain. All PINNs were trained with the following hyperparameters:

- optimizer : Adam then L-BFGS
- spatial samples : 256
- time samples : 100
- activation : $\tanh$

For the plain PINN experiments our architecture is a MLP with layers sizes as follows: [2, 128, 128, 128, 128, 1]. Boundary and initial conditions were enforced through a penalty in the loss function.

For the Causal PINNs we use the architecture described in the original paper, with periodic embedding and "modified MLP layers." The layer sizes were as follows: [`periodic embedding`, 128, 128, 128, 128, 1]. Our implementation uses much of the original code provided in [51]. We used only one time-window for training as this is what was chosen in [51] for the Allen-Cahn equation. The tolerance hyperparameter, which controls the degree to which the loss function enforces causal training, was set to 100 and 50 for the Allen-Cahn and Burgers' equations respectively. Additionally the $\lambda_{ic}$ parameter, which controls the loss functions' weighting on the initial condition, was set to 100 and 1 for the Allen-Cahn and Burgers' equations respectively. These were both chosen via a hyperparameter search. For details on the context and meaning of these hyperparameters see the original paper [51].

The PINNs trained for Allen-Cahn used 1000 steps of Adam followed by 30000 steps of L-BFGS. For Burgers' equations we used 1000 steps of Adam followed by 60000 steps of L-BFGS. More steps are needed for Burgers' equation in order to sufficiently resolve the sharp gradient in the solution due to the low viscosity number. We similarly choose a smaller timestep for Burgers' in the Neural Galerkin schemes. We note that fewer optimization iterations can be used for training PINNs but this resulted in much larger errors in our experiments. In any case, widely varying the number optimization iterations and other PINNs hyperparameters did not achieve errors in the range of what RSNG achieves in these examples.

