# OpenReview forum: "Randomized Sparse Neural Galerkin Schemes for Solving Evolution Equations with Deep Networks"
_NeurIPS.cc/2023/Conference — NeurIPS 2023 spotlight_

### Official Review · Reviewer_YG46 · 2023-07-03

**Soundness:** 3 good
**Presentation:** 3 good
**Contribution:** 3 good
**Rating:** 6
**Confidence:** 5

**Summary:**

The paper proposes a so-called randomized sparse neural galerkin (RSNG) algorithm to solve time-dependent PDEs. The basic idea is to
consider parameterized functions as ansatz spaces, with parameters that vary in time. Consequently, the method is of the sequential-in-time learning algorithms rather than global-in-time learning algorithms such as plain vanilla PINNs. Computing PDE residuals with this class of ansatz functions and collocating the loss (at every time instant) naturally leads to a least-squares type linear algebra problem which can be solved either directly or indirectly and has been previously considered. The author identify problems with this approach and propose a novel remedy by considering randomized sparse instantiations of the projection, loosely along lines of dropout, but in a more L2 sense. The algorithm is illustrated with some benchmark examples for PDEs.

**Strengths:**

1. The premise of the paper is interesting. It is well-known that global-in-time methods such as PINNs are ill-conditioned and hard to train and the paper investigates a method of potentially avoiding some of the pitfalls of global-in-time methods.

2. Randomized sparse algorithms can be efficient in the context described by the authors as long as certain conditions, such as incoherence that they loosely identify, are fulfilled.

3. Numerical experiments clearly demonstrate that RNSG outperforms the dense least-squares method as well as baselines such as plain vanilla PINNs considerably.

**Weaknesses:**

1. Clarity: The paper could do with a rewrite where authors elaborate on the rather succinct notation, for instance in the appendix. In particular, to aid the reader, they could write out the neural networks in line 81, for instance for a simple one-hidden layer shallow neural network, as to how the time variable and the space variable exactly enter into Eqn (2) -- the reader will clearly see the distinct roles played by time here. Similarly, further rationale could be provided for the choice n >> p in deriving Eqn (4), one could for instance refer to it a quadrature for Eqn. (3). Caption of Figure 2 could be better explained, in particular what is refit in Figure 2(b) and what does co-adaptation mean and how does Figure 2 (c) shows this property. Quite a lot of heuristics is provided in a hand wavy manner that is potentially inscrutable to a reader -- for instance, what does the lines 173-176 mean ?

2. Theory or rather the lack of it: The author do not attempt to provide any theoretical justification of their method. Some phrases are included here and there (e.g. line 162, lines 173-179 etc) but the reader feels that the authors know more about the theory then they present here. A discussion on these aspects could be illuminating. In particular, when is the method supposed to work and when not ? The lack of presentation of a theoretical perspective forces this reviewer to judge and criticize the paper on its empirical content as I proceed to do below.

3. Empirical evaluation: The authors consider three time-dependent PDEs i.e., 1-d Allen-Cahn, 1-d viscous Burgers with a relatively low viscosity and a 2-d Vlasov-Poisson with a given potential and on a Cartesian domain. As such the problems are fairly well-studied benchmarks and the authors clearly show that their RNSG algorithm handily outperforms its dense foundation and an iterative version. However, the following limitations of the experiments can be identified:

      a. The authors claim that it takes 18 mins for a PINN to obtain 6% errors for Allen-Cahn and almost 2 hours to obtain below 1% error for the simple viscous-Burgers example. This is unacceptably large and could very well be due to the use of ADAM as the optimizer. It is standard in PINNs literature to either use L-BFGS or start with ADAM and then switch to L-BFGS after say Order of 1000 iterations or so. This reviewer is curious if doing so reduces the run-time with PINNs ? In any case, the authors obtain significantly lower errors with RNSG -- so the timing is worth investigating.

   b. Clearly, PINNs and even Casual PINNs (I am not convinced that it takes more than 6 and half hours to solve Burgers' equation with casual PINNs) are not the best conditioned to be the only baselines here. Rather I would suggest the following baselines:

             b1) A method of lines version of PINNs (see the Raissi et al original PINNs paper) where the time is discretized with a RK4 solver and space with PINNs -- this is a fairer comparison as time is now iteratively updated with the PINNs and should be compared against your method, at least for some experiments.

            b2) Alternatives to PINNs such as ODILs (Karnakov et al https://arxiv.org/abs/2205.04611) which directly collocate the PDE residual (with an implicit time-discretization) appear to be much better conditioned and could be a strong baseline for your method.

         b3) The whole rationale would be out-compete standard numerical methods. Hence, it would be instructive to consider the computational cost of your ground truth generator. How much computational time does it take with this ground truth generator -- for all the experiments -- to reach comparable accuracy. Given that you require 13 mins to solve 1-D Allen-Cahn and 40 minutes to solve 1-D Burgers, traditional numerical methods would solve both these problems in a matter of secs, if not faster -- so where is the rationale for the use of your method ? One could argue that its rationale would be high-dimensional problems or complex geometries, which you don't consider empirically here.

4. Scope: In its current version, it is unclear how this algorithm scales with spatial (or parametric) dimension -- is it fairly dimension independent ? If so, where is the evidence for that -- for instance, what is the computational time for the 2-D Vlasov-Poisson problem ?

Another issue pertaining to the scope is that the method relies on an explicit time-stepping integrator such as RK4 -- can it be used for stiff problems, with implicit time stepping ? Otherwise, you will require too many iterations in time, something that PINNs and other global-in-time methods don't have to. It would be useful to see if your algorithm can work with large time steps for such stiff problems.

5. Comparison: The authors say that other works such as [1,7,14] have considered this approach and yet do not provide a clear comparison to these works. It is essential to distinguish between them and the current paper, in a related work section.

**Questions:**

In addition to the questions above in the "weaknesses" section -- the following minor ones should be addressed:

1. What is refit in Figure 2(b) -- it is also used in other figures ?

2. What is computational cost of RNSG for 2-D Vlasov-Poisson ? How does it compare to PINNs, especially when they are optimized with L-BFGS.

3. How do the results, particularly the choice of the sparsity parameter $s$ change with the viscosity $\nu$ for viscous-Burgers' ? How low can $\nu$ be to still allow for accurate approximation ?

4. Please mention the value of $s$ in each of the figures (for instance in the option or the Appendix) where you have not mentioned the value.

**Limitations:**

It is a mostly a reiteration of the section on Weaknesses:

-- No theoretical underpinning of the algorithm.

-- Lack of a clear description of the limitations of the algorithm.

--- How are the collocation points $x_i$ chosen ? The authors refer to a prior work but do not contextualize it here.

--- More baselines need to be evaluated -- in particular, the runtime of traditional numerical methods has to be mentioned to provide the reader with an idea of what is the potential pay-off with their method, if any ?

The basic idea is interesting and I am happy to reconsider my score if the authors address my questions appropriately.

---

> ### Author Rebuttal · Authors · 2023-08-08
>
> We thank the reviewer for their comments and detailed reading of our paper.
>
> > [...] where authors elaborate on the rather succinct notation, for instance in the appendix. [...]
>
> The reviewer mentions several good points on how to further improve the clarity of the paper. We will address these in the camera-ready version, if this paper gets accepted. In particular, we will carefully revisit the captions of Figure 2.
>
> > [...] The authors do not attempt to provide any theoretical justification of their method. [...] A discussion on these aspects could be illuminating. [...]
>
> We provide several theoretical justifications in the paper that show that our approach is principled rather than heuristic: (a) One of the theoretic underpinnings of our approach is random matrix sketching, as we briefly mention on lines 173-179. There is a robust literature which we cite [13,19] on matrix sketching that provides tight bounds and when and why these randomized strategies will be successful. We mention some of the key findings for us in lines 173-179. (b) We also mention "dropout" and "co-adaptation" as building blocks for our approach, which have been studied by the ML community in theory and practice and we give several references to it [3,9,43,45]. We agree with the reviewer that a comprehensive theory would strengthen the paper and we consider this as future work.
>
> > The authors claim that it takes 18 mins [..] use L-BFGS or start with ADAM and then switch to L-BFGS [...]
>
> We aim to provide a fair comparison by basing our experiments with PINNs and CausalPINNs on parameters that are used in the literature. For example, in case of CausalPINNs, we follow closely the original CausalPINNs paper (doi: https://doi.org/10.48550/arXiv.2203.07404) and use the implementation (link: github.com/PredictiveIntelligenceLab/CausalPINNs ) they provide which only uses the ADAM optimizer without L-BFGS. Our accuracy results are in alignment with what is reported in the original CausalPINNs paper using a similar number of ADAM iterations.
>
> We ran the experiment suggested by the reviewer of training PINNs with 1000 ADAM iterations followed by L-BFGS. We got lower runtimes (e.g,  for Allen-Cahn 841s, for Burgers' 3451s) but with a slightly higher error. Our RSNG remained faster in terms of runtime and superior in terms of accuracy (2-3 orders of magnitude more accurate). Furthermore, there are also several steps to further tune the sparsity parameter s of RSNG to improve runtime performance without sacrificing too much accuracy of RSNG. We will incorporate these experiments and new benchmarks into the paper if accepted.
>
> > [...] I would suggest the following baselines [...]
>
> The reviewer suggests we look at the method of line PINNs (MOL PINNs) as a baseline comparison. In order to accurately reproduce this method we take the implementation from the original paper given (https://github.com/maziarraissi/PINNs) and swap in our problem setups for Allen-Cahn and Burgers.
>
> For Allen-Cahn the default hyperparameters result in higher accuracy (error 2.54e-03) but slower runtime (runtime 1247s) than our PINNs experiment. All results remain below the performance of RSNG (error 1.66e-05, runtime 776s) and thus support the overall conclusion of the paper. For Burgers, in our experiment, the method fails to converge (error 8.81e-01; runtime 1386s). This was consistent across a brief hyperparameter sweep. From our examination this could be due to our benchmark using a lower viscosity term than the problem setup in the original paper.
>
> In terms of comparison to classical numerical solvers: the aim of the paper really is demonstrating superiority over dense Neural Galerkin schemes, for which we think the benchmark problems that we consider are sufficient. Settings where Neural Galerkin schemes outperform classical methods (e.g., high-dimensional PDEs) come with their own set of challenges that are addressed by other references (e.g., [7]) and go beyond the scope of this paper. Although we provide a proof-of-concept for high-dimensional PDEs (see below). We also clearly mention this in the Limitations section; see also response to reviewer mkqi.
>
> > What is the computational time for the 2-D Vlasov-Poisson problem?
>
> We add plots showing the computational time for 2D Vlasov in our global response Fig 11. We obtain a speedup of 10x compared to the dense method.
>
> > ​​[How does the] algorithm scales with spatial (or parametric) dimension [...] ?
>
> The runtime of RSNG is dominated by the number s of network weights that are updated in each time step. How small we can chose s is determined by the singular value decay of the Jacobian. To show this can be independent of spatial dimension, we fit neural network to the solution of a five-dimensional Fokker-Planck equation; see Fig 14 in the global response PDF. The singular values of the Jacobian decay quickly as well in this case; indicating that in high dimensions choosing s << p is valid.
>
> > [...] the method relies on an explicit time-stepping integrator such as RK4 -- can it be used for stiff problems, with implicit time stepping [...]
>
> Neural Galerkin schemes can be formulated with implicit time-stepping schemes, which is done, for example, in Reference [7]. So all what we propose applies just as well to implicit time-integration schemes.
>
> > Comparison: The authors say that other works such as [1,7,14] have considered this approach and yet do not provide a clear comparison to these works. [...]
>
> We do provide comparisons: All of these works are performing dense updates. So we do provide numerical comparisons, which we label as ``dense’’ in our numerical experiments. We will further clarify this in the camera ready version, if accepted.
>
> > How are the collocation points chosen?
>
> We detail in the appendix that we select our points on an equispaced grid in the spatial domain, to avoid sampling error polluting the overall results.

---

> > ### Comment · Reviewer_YG46 · 2023-08-17
> > **Reply to the authors' rebuttal.**
> >
> > At the outset, I would like to thank the authors for their rebuttal and apologize for my delay in responding to it. The authors have addressed some of my concerns but quite a few still remain as are outlined below:
> >
> > 1. Regarding clarity: The authors promise to clarify several issues in the CRV but I would appreciate answers to the specific questions that I have raised in point 1 (Weakness) of my original review.
> >
> > 2. Providing 5-6 lines of pointers to references does not constitute any substantial theory in my view. On the other hand, I understand that the authors will pursue theoretical questions in a future work
> >
> > 3. This reviewer is not convinced by the PINNs baselines. I ran plain vanilla PINNs with the viscous Burgers equation with L-BFGS (with similar viscosity as the authors) and obtained around 1% relative error in less than 10 secs without much hyper parameter tuning. So, I find it hard to believe that the best that PINNs can do is to take 30 mins to solve the 1-D Burgers equation. By no means I am saying that the authors intentionally provided weak baselines. Far from it but it will convince the readers more if the strongest baselines are used. In any case, I fully believe that your method will be more accurate than PINNs.
> >
> > 4. I come to my main criticism of the paper which the authors have not addressed well. PINNs or your algorithms are not going to be competitive with respect to traditional numerical methods in the regimes that your numerical examples are set. Standard time-stepping finite difference schemes will readily (by orders of magnitude in both accuracy and run-time) outcompete these algorithms. This point has to be made in the limitations (as another reviewer has also acknowledged). However, this does not mean that these methods have no utility -- their utility will show up in parametric PDEs (with lots of parameters) and inverse problems. These motivated my question about how your method scales with parametric dimension. I am not satisfied with the current answer -- an easy fix would be have been to solve a genuinely parametric problem and see how your method scales.
> >
> > I look forward to the authors' response to my questions and apologize again for the delay in replying.

---

> > > ### Author Response · Authors · 2023-08-20
> > > **Response to Comment by YG46**
> > >
> > > We thank the reviewer for their continued comments and close reading of our paper.
> > >
> > > > [...] I would appreciate answers to the specific questions that I have raised in point 1 [...]
> > >
> > > We followed guidelines and kept the response short, but here a few more specific steps that we will make:
> > >
> > > 1) Write out explicitly how time and space enter a simple one-hidden layer shallow neural network.
> > > 2) Clarify the choice n >> p is to ensure good estimates of the gradients.
> > > 3) For Fig 2(b) we will emphasize standard regression on ground truth data (refit) will find better regions in the parameter space than dense Neural Galerkin schemes.
> > > 4) For Fig 2(c) we will clarify that co-adaptation means that the gradient with respect to different parameters are highly correlated as depicted in Fig 2(c).
> > >
> > > > [...] plain vanilla PINNs [...] (with similar viscosity as the authors) and obtained around 1% relative error in less than 10 secs [...] In any case, I fully believe that your method will be more accurate than PINNs.
> > >
> > > We choose hyperparameters (documented in appendix) for PINNs and RSNG for maximizing accuracy rather than runtime, as indeed this is our focus as the reviewer agrees that this is the strength of the method ("fully believe that your method will be more accurate than PINNs"). In any case, we will definitely discuss the reviewer’s important point that PINNs can run fast at a low accuracy in the camera-ready version, if accepted. We also note that from Fig 5(a) we see that we have ample room to tune the sparsity s of our RSNG to get lower runtimes. A brief experiment shows, if we tune hyperparameters of RSNG towards speedup rather than accuracy, then RSNG solves Burgers’ equation in about 10s with around 1% relative error.
> > >
> > > The reviewer’s point is well taken and we will be happy to add an additional plot that more explicitly shows the accuracy/speed trade-off for PINNs and RSNG (similar to Fig 4b). This will convey the reviewer’s point that PINNs can run fast at a low accuracy. Overall our experiments indicate that this plot will be consistent with the message we intend to convey with Table 1.
> > >
> > > > PINNs or your algorithms are not going to be competitive with respect to traditional numerical methods [...]
> > >
> > > There are many applications in which Neural Galerkin schemes succeed where traditional numerical methods struggle: (a) high-dimensional PDEs, (b) model reduction of transport-dominated problems, (c) domains with complex geometries, and others. We agree with the reviewer that parametric PDEs are another important possible application. These are all areas of active research.
> > > We emphasize that this paper is of interest to the broader community as RSNG has the potential to improve speed and accuracy of the Neural Galerkin scheme on all the above applications. Our goal is to convincingly show the superiority of RSNG over dense Neural Galerkin schemes which, as of now, are limited by computational costs and overfitting. We believe this is a reasonable scope for our paper. We have provided strong evidence that RSNG is a promising method which future researchers can adopt or adapt to their specific application of interest.
> > >
> > > >  This point has to be made in the limitations [...]
> > >
> > > This point was made on line 317, but we will be sure to elaborate on it in the camera ready version, if accepted, according to discussions we had here.

---

> > > > ### Comment · Reviewer_YG46 · 2023-08-21
> > > > **Reply to the authors**
> > > >
> > > > I thank the authors for their reply. I believe that if they make the changes in a CRV, it will improve the scope of the paper. I still maintain my main point of criticism: I agree with the authors that neural galerkin schemes (just like PINNs, Deep ritz and variants) have applications in very high-dimensional problems, inverse problems, parametric problems etc. My concern is that you do not demonstrate how RSNG performs in these critical problems, which will be the key to its widespread acceptance in the community. The point about RSNG being better than Neural Galerkin is well-made but its wider applicability has not been demonstrated. Nevertheless, this is a point that can be investigated in future research. I raise my rating accordingly.

---

> > > > > ### Author Response · Authors · 2023-08-21
> > > > >
> > > > > We thank the reviewer for the insightful comments and the good discussion that will help improve the camera ready version, if accepted.

---

### Official Review · Reviewer_mkqi · 2023-07-06

**Soundness:** 3 good
**Presentation:** 2 fair
**Contribution:** 3 good
**Rating:** 7
**Confidence:** 4

**Summary:**

The paper suggests a simple but effective way to train sequential-in-time models by randomly sampling weights to update. The method avoids overfitting and realizes speedup because only a small part of parameters is updated. The experiments show that the proposed model offers the best performance in terms of speed and accuracy.

**Strengths:**

* Although the method is simple, the idea is based on the reasonable assumption that parameters are redundant locally in time. Due to its redundancy, the Jacobian matrix is prone to be singular or stiff and takes a huge computational budget for training with the existing sequential-in-time methods. The proposed method elegantly addresses these issues by randomly sampling weights to update.
* The experiments clearly demonstrate that the proposed method works efficiently in terms of computation time and accuracy compared to the considered baseline methods.

**Weaknesses:**

* Figure 5 (b) and (c) suggest that sampling could lead to instability (see also Questions). The reviewer recommends that the authors show predicted solutions in figures to assess the smoothness of the prediction. Sometimes smoothness is key because some downstream tasks may perform additional differentiation on the prediction.

Minor points:
* In the first equation in Equation (1), the domain of t should be defined in the open set (see, e.g., Jürgen Jost "Partial Differential Equations ThirdEdition" (2012) Chapter 1) to make differentiation possible everywhere in the domain. That's why we need the initial condition.

**Questions:**

* Is the method faster than classical numerical solvers (e.g., finite difference method)? It could be important to compare with the classical methods.
* Figure 5 (b) and (c) suggest that sampling could lead to instability. What is the reason, and any ideas to address this? The reviewer guesses that the updated weights are not smooth in time, i.e., the indices of the weight matrix are drastically changed at a time $t$ and the next time $t + \Delta t$. Could it be interesting to introduce some "momentum" to make the change of indices moderate in time?

**Limitations:**

* The authors addressed limitations. Another limitation could be handling boundary conditions other than periodic (e.g., Dirichlet and Neumann).

---

> ### Author Rebuttal · Authors · 2023-08-08
>
> We thank the reviewer for their comments and detailed reading of our paper.
>
> > [...] reviewer recommends that authors show predicted solutions in figures to assess the smoothness of the prediction. Sometimes smoothness is key because some downstream tasks may perform additional differentiation on the prediction.
>
> We do not observe that the random sampling is affecting the smoothness of the solution field. Changes in the weights are smoothed out by the network parametrization and its regularity. Furthermore, the sequential-in-time training is performed with a regular time integration scheme, which also leads to smooth updates in classical numerical solvers. To help readers assess the smoothness, we plot the predicted solution at various time steps (see global response PDF). We will add these figures to the appendix of the camera ready version, if accepted.
>
> > Figure 5 (b) and (c) suggest that sampling could lead to instability. What is the reason, and any ideas to address this? The reviewer guesses that the updated weights are not smooth in time, i.e., the indices of the weight matrix are drastically changed at a time t and the next time t + \Delta t. Could it be interesting to introduce some "momentum" to make the change of indices moderate in time?
>
> The reviewer raises an interesting point: The residual can be less smooth in time because at each time step a different random sketch of the Jacobian is used for approximations. We note two things: first, the residual is a nonlinear function of the solution and thus can be rough whereas the solution remains smooth in time (see comment to previous question). Second, we do not see empirically that the rougher residual is an issue for stability in our experiments. However, we certainly very much appreciate this really insightful comment. Indeed, it would be interesting how different sampling schemes (e.g., leverage score sampling to improve the random sketch) and adding momentum (as proposed by the reviewer) influences the residual. We will add a discussion in the camera-ready version (if accepted) that this is an important point to study..
>
> > Is the method faster than classical numerical solvers (e.g., finite difference method)? It could be important to compare with the classical methods.
>
> In this work, we focus on two pervasive challenges (high computational costs, overfitting) of Neural Galerkin schemes (and sequential-in-time methods in general) and demonstrate on simple benchmark problems the superiority of our RSNG approach over regular (dense) Neural Galerkin schemes. Because the benchmarks are standard problems (see the section on Limitations, where we make this very clear), it is unlikely that Neural Galerkin schemes achieve much speedup over classical numerical methods for these problems. However, there are several other settings where Neural Galerkin schemes are beneficial compared to traditional numerical solvers with linear parametrizations, e.g., (a) high-dimensional PDEs, (b) model reduction of transport-dominated problems (slow decay of n-width), and (c) domains with complex geometries. Yet, each of these areas poses its own unique set of challenges that are active research and go far beyond the scope of this paper, in our opinion. This is why we demonstrated superiority of our RSNG approach over regular (dense) Neural Galerkin schemes on simpler benchmark problems.
>
> To further address the this, we added a proof-of-concept experiment that indicates that RSNG can be applied to high-dimensional PDEs, a problem domain where classical numerical methods would fail. We fitted networks to solutions of five-dimensional Fokker-Planck equations and plot the decay of the singular values of the Jacobian (see Figure 14 in global response PDF). The decay is similarly fast as in our benchmark examples in the paper, which is an indication that RSNG can be successfully applied in such a high-dimensional setting as well.
>
> > [...] handling boundary conditions other than periodic (e.g. Dirichlet and Neumann).
>
> We will add comments in the camera-ready version (if accepted) to emphasize that (a) Dirichlet boundary conditions typically can be imposed in the network architecture and (b) Neumann boundary conditions (as well as Dirichlet) can be imposed via constraints, which is common when numerically solving PDEs with neural-network parametrizations.
>
> > [...] the domain of t should be defined in the open set [...]
>
> We will make this correction. This is a good catch and improves the technical quality of the paper.

---

> > ### Comment · Reviewer_mkqi · 2023-08-16
> >
> > Thank you for the response. It addresses the concerns raised by the reviewer.
> >
> > > it is unlikely that Neural Galerkin schemes achieve much speedup over classical numerical methods for these problems.
> >
> > This point could also be added in section 5 to clarify the scope and limitations of the work.

---

> > > ### Author Response · Authors · 2023-08-17
> > > **Response to Comment by mkqi**
> > >
> > > We thank the reviewer for their time and accurate assessment of our paper. While we touched on this point on line 317, we will be sure to elaborate on it in the camera ready version, if accepted, according to discussions we had here. We are glad we were able to address all the reviewer's concerns.

---

### Official Review · Reviewer_8WGb · 2023-07-06

**Soundness:** 3 good
**Presentation:** 3 good
**Contribution:** 3 good
**Rating:** 7
**Confidence:** 2

**Summary:**

The authors suggest an alternative optimization scheme for Neural Galerkin methods - a family of methods in which parameters of a neural network are optimized so that the gradients of the solution obey a given PDE with the notable quality being that rather than solving over all of space-time, the parameters of the network evolve in time to match the system evolution equation - in which only a subset of the surrogate model's parameters are updated at each time step. The author's show that this approach seems to lead to improved accuracy at a fixed FLOP budget and improved speed to a fixed accuracy threshold.

**Strengths:**

- The paper is well written and interesting.
- Jargon is generally defined shortly after it is used.
- As someone who is not very familiar with Neural Galerkin methods, it seemed that most questions that immediately came to mind after reading segments were answered quickly.
- Overall, the paper was clear, compelling, and all claims were well-supported by the provided evidence.


**Weaknesses:**

The main weakness of the paper is that while the paper is clearly providing value to a current research topic, it could do a better job of explaining where Neural Galerkin methods are (or could be) practically useful. The PINN comparison is enough for me to be comfortable recommending approval, but given the method requires knowledge of $f$, it seems like true comparison would be numerical solvers which, as the authors mention in the conclusion, are still quite a bit ahead of the PINN comparisons on these problems. Without that, it's difficult to see whether the paper would be interesting to a wider audience.

Line-item Issues:
- Figure 2 is not very clear. a) makes sense, but (b) could use a better captioning so that it can be understood independently of the text and (c) is unclear even after reading the text.
- Figures 3, 4 are similar in that the caption is currently insufficient to explain the figure.


**Questions:**

- Were the PINN comparisons also performed with rational activation functions?
- How do these methods compare against numerical methods?
- Given the conclusion implies the comparison is unfavorable, it would be very valuable to see examples from the set of problems where numerical methods are expected to do poorly.

**Limitations:**

Limitations are well covered and no ethical issues are foreseen.

---

> ### Author Rebuttal · Authors · 2023-08-08
>
> We thank the reviewer for their comments and detailed reading of our paper.
>
> > Were the PINN comparisons also performed with rational activation functions?
>
> Tanh activations are usually a standard choice in the PINN literature, which is what we used too. However, to ensure the choice of the activation function was not a key part of our advantage we reran our PINN experiments with rational activation functions. The results are comparable in terms of error, but for additional computation cost due the additional parameters. Allen Cahn: 6.97e-2 (cmp 6.42e-02 with tanh activations as in paper). Burgers: 7.04e-4 (cmp 1.32-e3 with tanh activations as in paper). Thus, regardless of the choice of activation function, RSNG still achieves a higher accuracy of 1 to 3 orders of magnitude, and lower runtime.
>
> > [...] while the paper is clearly providing value to a current research topic, it could do a better job of explaining where Neural Galerkin methods are (or could be) practically useful. [...] How do these methods compare against numerical methods? Given the conclusion implies the comparison is unfavorable, it would be very valuable to see examples from the set of problems where numerical methods are expected to do poorly.
>
> We briefly mention the application scenarios in the conclusions but agree with the reviewer that this should be brought up more prominently: (a) high-dimensional PDEs, (b) model reduction of transport-dominated problems (slow decay of n-width), (c) domains with complex geometries, and others. In this paper, our main goal is to show superiority over the regular (dense) Neural Galerkin schemes for which we focus on simple benchmarks (see Limitations section) as we think these are sufficient for demonstrating the benefits of our RSNG. It is our hope that our RSNG approach is a reasonable step forward towards tackling current limitations of Neural Galerkin schemes to be a building block for fruitful approaches for these other applications, as we mention in Conclusions and Limitations sections.
>
> To further address the reviewer’s concern, we added a proof-of-concept experiment that indicates that RSNG can be applied to high-dimensional PDEs as well: We fitted networks to solutions of five-dimensional Fokker-Planck equations and plot the decay of the singular values of the Jacobian (see Figure 14 in global response PDF and reply to comment by reviewer Gbtz). The decay is similarly fast as in our benchmark examples in the paper, which is an indication that RSNG can be successfully applied in such a high-dimensional setting as well.
>
> > [Caption of] Figure 2 is not very clear. [...]
>
> We agree the captions could be made more clear such that they stand independent of the main text: For Fig 2(b) we will emphasize standard regression on ground truth data (fit) finds better regions in the parameter space than dense (in contrast to our RSNG) Neural Galerkin schemes. We will similarly clarify the other captions mentioned by the reviewer.

---

> > ### Comment · Reviewer_8WGb · 2023-08-13
> >
> > Thank you authors for your response. I am satisfied by the responses to points 1 and 3. I feel the high-dimensional PDEs/complex geometry/ROM argument is still a hypothetical and am not convinced that the advantage over classical methods would actually materialize without experiments. However, at this point there is a significant body of work in the literature with this issue, and I agree with the authors' position that they are showing an efficiency gain over what currently exists and feel comfortable confirming my current recommendation of acceptance on the basis of that improvement and the high quality work presented in the paper.

---

> > > ### Author Response · Authors · 2023-08-17
> > > **Response to Comment by 8WGb**
> > >
> > > We thank the reviewer for their time and high quality comments on our paper.
> > >
> > > With regards to applications in high-dimensional PDEs, complex geometry, and ROMs we would like to highlight the following very recent papers (after our submission) which apply Neural Galerkin Schemes on these problems. We believe that RSNG is a powerful tool to improve the viability of these methods:
> > >
> > > [High Dimensional] Wen, Yuxiao, Eric Vanden-Eijnden, and Benjamin Peherstorfer. "Coupling parameter and particle dynamics for adaptive sampling in Neural Galerkin schemes." arXiv preprint arXiv:2306.15630 (2023).
> > >
> > > [Complex Geometry] Kast, Mariella, and Jan S. Hesthaven. "Positional Embeddings for Solving PDEs with Evolutional Deep Neural Networks." arXiv preprint arXiv:2308.03461 (2023).

---

### Official Review · Reviewer_LKD5 · 2023-07-06

**Soundness:** 3 good
**Presentation:** 4 excellent
**Contribution:** 3 good
**Rating:** 7
**Confidence:** 4

**Summary:**

In this paper, the authors propose a modified Neural Galerkin scheme for the solution of time-dependent PDEs, aiming at reducing computational cost and improving accuracy by avoiding overfitting.

The target solution $u(x,t)$ is approximated with a nonlinear parameterisation $\theta(t)$, so that $u(x,t)\approx \hat{u}(x;\theta(t))$ (this can be interpreted as a nonlinear form of reduced-order modelling). In practice, $\theta(t)$ represents the parameters of a NN responsible for outputting $\hat{u}(x;\theta(t))$ for a given $x$. At each time-step $t$, the goal is to train the network to output the solution at the next time-step. As is classical with Galerkin methods (and derivative thereof), this is achieved by following a variational principle, whereby the norm of the residual of the PDE is minimised. This can be recast as a least-square minimisation problem in the unknown $\dot\theta$: once solved, the gradient of $\theta$ is then used to guide the update of the NN.

The novel contribution of the paper lies in applying sketching to simplify the solution of this least-square problem. This is rooted in the observations that:
- Often, the target matrix in the system $J(\theta)$ (a collation of the gradients of $u$ wrt $\theta$, sampled at specific collocation points) is low-rank, implying redundancy in the components of $\theta$
- Updating the whole $\theta$ tends to overfit, which in turn leads to accumulation of errors and poor quality of the solution as time evolves.
This justifies reducing the space of parameters to update, and has the boosted effect of reducing computational time due to the effective shrinking of the least-square problem.

The performance of the novel method is verified on three benchmark PDEs (two 1D and one 2D), and expressed in terms of runtime and norm of residual. The proposed method outperforms both its dense (non-sketched) counterpart, and the global-in-time method (Causual) PINN.

**Strengths:**

- The sketching simplification at the core of the method is well-justified, also with experimental results. Moreover, I particularly appreciate the fact that the intuition behind using sketching is rooted both in algebraic considerations (pertaining rank of J), but is also interpreted through the lenses of regularisation techniques in NN (dropout). Both views are valid, and contribute to supporting the hypothesis
- The paper is well written, neatly structured, with the right level of rigour and a good introduction to the topic

**Weaknesses:**

- Possibly the breadth of the contribution: after all, it represents a relatively small modification to an already existing method, and the modification itself takes inspiration from already-established techniques. Still, I appreciate this work, and the contribution seems valid and useful
- Some experiments might be expanded to further substantiate the claims (see questions below), but this should be straightforward to address

**Questions:**

- How does Fig2a look like for Burgers’ and Vlasov? And if we vary the network size? I’m trying to understand whether the low-rankness observation is affected somehow by the type of problem/model considered, or if it is robust to these variables (Fig5a partially hints at this, but it would draw a fuller picture to include comparisons throughout the paper)
- Line 257: can you report the actual value of the error in this case? Also, how does Fig4(a) look like for Allen-Cahn and Vlasov? This builds up on the previous question: I’m trying to understand how robust these findings are wrt problem chosen
- Fig 4(b): how does time complexity scale with $s$? In my mind, this was supposed to be the bottleneck of the algorithm, and if you’re using a direct solver to solve the associated least-square system, complexity (and time) should scale quite heavily in $s$ (you mention "quadratically" a couple of times: what exactly are you using as solver?). But by eyeballing the datapoints in the plot, it looks almost linear? Am I missing something?
- I liked Fig5(a): the fact that it’s U-shaped really fits the overfitting assumption (pun intended), and I think it really helps nailing down the validity of the method. I also like the fact that it shows that Vlasov kinda falls outside the trend - but this makes sense: it’s a more complex problem after all (if anything, because it’s 2D). Ultimately, I expect these graphs to be both problem- and NN- dependent: if you had chosen a NN with more parameters, likely at a certain point it would've started overfitting again, even for Vlasov. In any case, it’s extremely useful to know when a method might start breaking down, and it should be pointed out!

Minor corrections
- Fig3(d):
    - For the x-label: I reckon $757$ is the fixed value of $s$? Just write $p (s=757)$ or something on those lines. Same for Fig5
    - Please report an indication of scale (for example, by explicitly specifying a numerical value for the “Dense” tick)
    - I reckon the relative error here refers to the end time $T=4$?
    - Maybe they would look better if the figures shared the same scale for the y-axis? This way one could align them and compare better among them? Not sure if they’d get too squashed though
- Line 237: I can see error bars only for Fig3(d). If for Fig(a-c) they’re too small to see, please state it explicitly. If you’re not reporting them, why not? Similarly for Fig5
- Line 245: please clarify: “we plot the relative error [with respect to the underlying solution] over time”

**Limitations:**

The main limitations I could identify (particularly, dependency on properties of the target Jacobian) are addressed and made explicit in the corresponding section

---

> ### Author Rebuttal · Authors · 2023-08-08
>
> We thank the reviewer for their comments and detailed reading of our paper.
>
> > Some experiments might be expanded to further substantiate the claims (see questions below), but this should be straightforward to address
>
> We provide additional figures/experiments that all further support the claims of the paper:
> - added figures in global response PDF that show the singular value decays of the Jacobians corresponding to all equations  (Global response PDF Fig 10)
> - added figures in global response PDF that show additional details of the runtime speed-ups of RSNG for all equations (Global response PDF Fig 11 and 12)
> - proof of concept for high-dimensional problem (Global response PDF Fig 14)
>
> > How does Fig2a look like for Burgers’ and Vlasov? And if we vary the network size? I’m trying to understand whether the low-rankness observation is affected somehow by the type of problem/model considered, or if it is robust to these variables (Fig5a partially hints at this, but it would draw a fuller picture to include comparisons throughout the paper)
>
> The reviewer is completely right that the low-rankness is affected by the problem/model. We added Figure 10 (uploaded in the global response, and in camera-ready version if paper gets accepted) that show the decay of the singular values of the Jacobian for neural networks fit to Burgers' and Vlasov. For Burgers', the singular values decay similarly quickly as for the Allen-Cahn example. As the reviewer observes, and in agreement with our numerical experiments and Fig 5(a) in the paper, Vlasov is a more complex problem that exhibits a less sharp decay but is still distinctly rank deficient.
>
> We additionally demonstrate what happens when we vary the network size in global response PDF Fig 14. Here we fit a neural network to a numerical solution of the Fokker-Planck equation in 5 dimensions. The results show that as we increase the size of the network, the Jacobian has a steeper decay in the singular values so that random sketching strategies will likely be successful. We will add details of this experiment in the appendix in the camera-ready version, if this paper gets accepted.
>
> This agrees with and also addresses the reviewer's related comment:
>
> > that Vlasov kinda falls outside the trend - but this makes sense: it’s a more complex problem after all (if anything, because it’s 2D).
>
> Indeed, how sparse the updates can be, depends on the complexity of the problem, which is reasonable and expected and in alignment with the results of the additional figures mentioned in the previous paragraph. If the paper gets accepted, we will add this discussion to the appendix of the camera-ready version. We note, however, that low-rankness is only one part of what RSNG addresses. Additionally there is the overfitting problem, which we detail in the paper and which is addressed by the randomization.
>
> > Line 257: can you report the actual value of the error in this case? Also, how does Fig4(a) look like for Allen-Cahn and Vlasov? This builds up on the previous question: I’m trying to understand how robust these findings are wrt problem chosen
>
> The actual value of the error on line 257 is 2E-4.  We uploaded a version of Figure 4(a) for each of the numerical examples above and we report the error values in the caption (global response PDF Fig 11). We will add these to the appendix of the camera-ready version if accepted.
>
> > Fig 4(b): how does time complexity scale with s? In my mind, this was supposed to be the bottleneck of the algorithm, and if you’re using a direct solver to solve the associated least-square system, complexity (and time) should scale quite heavily in s (you mention "quadratically" a couple of times: what exactly are you using as solver?). But by eyeballing the data points in the plot, it looks almost linear? Am I missing something?
>
> We use a direct solver based on SVD for which costs scale quadratically in s. Please note that the aim of the method is to reduce s: Halving the number of parameters that need to be updated (halving s) with our RSNG approach vs dense Neural Galerkin leads to quadratic speedups of the solve step. We imply this on line 110, but we will make this more explicit in the camera-ready version, if accepted. Note that we also show speedups compared to dense Neural Galerkin with iterative least-squares solvers, which suffer from a condition problem, as we discuss in the paper.
>
> The quadratic scaling of the speedup is not directly reflected in Figure 4b because of the low values of s and other effects (e.g., Jacobian computations, parallelization on GPUs). We will add comments to clarify this in the camera-ready version if accepted. Additionally, we added a figure (global response PDF Fig 12) where we plot directly s vs solver runtime for each of our equations. These figures show more precisely the quadratic scaling in s. We plan to add these to the appendix, if accepted.
>
> > Line 237: I can see error bars only for Fig3(d)...
>
> The error bars are plotted but too small to be seen. We will state this explicitly in the camera-ready version, if accepted.
>
> > I reckon the relative error here refers to the end time
>
> The relative error is computed over the whole space-time domain. We will make this formula explicit in the appendix, if accepted.
>
> > Minor corrections [...]
>
> We plan to make all other minor corrections that the reviewer suggested.

---

> > ### Comment · Reviewer_LKD5 · 2023-08-11
> >
> > I thank the authors for addressing my concerns. I believe the additional results showcased really helped drawing a fuller picture of the capabilities / limitations of the proposed method, and as such contributed to increasing the overall quality of the paper. In light of the new results, I still have a few minor concerns that I hope can be promptly addressed:
> > - Fig10(c) in the additional pdf indeed shows that the low-rankness property is quite heavily related to the problem considered: Vlasov in particular shows a much slower decay than I was anticipating. Luckily, cross-checking the results in Fig5 with those in Fig10, it seems like covering singular values up to ~1e-4 suffices to get reasonable accuracy. Moreover, it’s good that increasing NN size seems to accelerate this decay (as per Fig14). Notice however that there’s going to be a trade-off: deeper NN makes the forward pass more expensive / increasing $s$ makes recovering the least-square solution more expensive. Please make sure to appropriately elaborate on this in the final version of the paper
> > - In Fig 14, I think reporting decay wrt the _absolute_ number of nodes considered (as opposed to a percentage) is much more indicative. “How large an $s$ do I actually need” is the question to ask here, not just “how quickly do the singular value decay”, since ultimately it is $s$ that defines the cost of the LS solve. How many nodes are you considering for this problem?
> > - You don’t fully address my remark on Fig5(a)? Given the time constraint, would it be feasible to extend the graph for Vlasov? After all, verifying that indeed the U-shape trend eventually occurs for this problem too would solidify much more your claim on the method being effective against overfitting

---

> > > ### Author Response · Authors · 2023-08-17
> > > **Response to Comment by LKD5**
> > >
> > > We thank the reviewer their time and we are happy that we were able to addressing their concerns. We agree through this discussion we have been able to improve the quality of the paper.
> > >
> > > With regards to the minor concerns,
> > >
> > > > Fig10(c) in the additional pdf […] Notice however that there’s going to be a trade-off: deeper NN makes the forward pass more expensive / increasing s makes recovering the least-square solution more expensive.
> > >
> > > The reviewer raises two important points that we will highlight in the camera-ready version if accepted: (a) Selecting s to truncate near singular values of 1e-04 seems sufficient in the Vlasov example, which shows that selecting a small s is fine even if the singular values decay not too fast. (b) We will discuss the trade-off between the costs of a forward pass and the sparsity parameter s. This is an important point.
> > >
> > > > In Fig 14, I think reporting decay wrt the absolute number of nodes considered [...]
> > >
> > > The number of nodes varies with the size of the network. We will additionally add this plot in the camera-ready version, if accepted.
> > >
> > > > You don’t fully address my remark on Fig5(a)? Given the time constraint, would it be feasible to extend the graph for Vlasov [...]
> > >
> > > We extended the graph with a larger network but the solution of the Vlasov problem is highly oscillatory (multi-scale) which introduces errors and these hide the u-shaped uptick, at least for network sizes that were tractable for us to compute. To show the effect is present on other 2D problems, we ran an experiment on a 2D linear advection PDE with a time varying coefficient. Here we observe the same U-shape seen in Burgers and Allen-Cahn in Fig 5(a) for the same sized network. We have error 3.57E-04 for s=1600/3460 (sparse) and 1.14E-03 for s=3460/3460 (dense). We are happy to add the details of the experiment to the camera ready version if accepted. We believe this solidifies our claim that RSNG ameliorates overfitting.
> > >
> > > As future research investigates more expressive or tailored architectures, it is likely overfitting will become an issue even for more complex problems with various error sources such as in the Vlasov problem. We believe we have shown ample evidence that when overfitting does occur, RSNG is a reliable way to ameliorate this problem. Lastly we reiterate that overfitting is only one part of what RSNG addresses—our method, even for Vlasov, also provides orders of magnitude speed up while sacrificing little to no accuracy.

---

> > > > ### Comment · Reviewer_LKD5 · 2023-08-18
> > > >
> > > > I thank the authors for their reply, with which I consider my concerns addressed. I'm reasonably satisfied with the current state of the paper, and I've increased the score accordingly.

---

> > > > > ### Author Response · Authors · 2023-08-20
> > > > >
> > > > > We are happy we were able to address the reviewers concerns. We appreciate the reviewer raising their score accordingly. We have certainly improved the quality of the paper from this discussion.

---

### Official Review · Reviewer_Gbtz · 2023-07-08

**Soundness:** 3 good
**Presentation:** 3 good
**Contribution:** 3 good
**Rating:** 8
**Confidence:** 3

**Summary:**

The paper introduces Neural Galerkin scheme to sparsely update neural network over the sequential-in-time training. In particular, the proposed method randomly update sparse subsets of network parameters at each time step. With randomized sparse updates, overfitting problem of sequential-in-time methods can be mitigated, leading to better accuracy at test times. Moreover, the computational costs of training is reduced without losing expressiveness of the model, while accelerating the training process. Proposed method is validated on a wide range of evolution equations.

**Strengths:**

* The paper is well organized and easy to follow. It also has a strong motivation inspired by the Neural Galerkin schemes and neural network dropouts.

* The paper has a strong theoretical grounds and corresponding toy examples to support the claim. For instance, two challenges of previous sequential-in-time training are well addressed in the section 2.2 with adequate examples.

* The paper presents a wide range of ablation studies and various experimental settings to show the effectiveness of the proposed method. Most of the claimed benefits of the proposed method are validated empirically, including lower error rates and faster running time.

* The proposed method is considered a big contribution for the field of sequential-in-time training methods.

**Weaknesses:**

* The performance of the model is only validated in standard benchmark equations. Validation in high-dimensional PDE problems can benefit the paper.

**Questions:**

* The proposed method is inspired by dropout in neural networks. Thus, I am curious about comparison results with PINN and CausalPINN, with different rates of dropouts applied.

**Limitations:**

Limitations are addressed by the authors

---

> ### Author Rebuttal · Authors · 2023-08-08
>
> We thank the reviewer for their comments and detailed reading of our paper.
>
> > The proposed method is inspired by dropout in neural networks. Thus, I am curious about comparison results with PINN and CausalPINN, with different rates of dropouts applied.
>
> To address the reviewer’s point, we performed numerical experiments with PINNs and dropout. We can report that for various levels of dropout, the error of PINNs increases compared to no dropout as in our experiments in the paper. We believe this is because for PINNs the train and test distribution are nearly identical, therefore it is not beneficial to increase training loss for lower test loss as is accomplished by dropout.
>
> > [...] Validation in high-dimensional PDE problems can benefit the paper.
>
> While not being a comprehensive numerical experiment with high-dimensional PDEs, we conducted the following proof of concept and added a figure (global response PDF, Fig 14): We fitted a neural network to a numerical solution of the Fokker-Planck equation in 5 dimensions. The results show that for a sufficiently large network, the Jacobian has a low-rank structure as in the examples in the paper and a steep decay in the singular values so that random sketching strategies will likely be successful. We will add details of this experiment in the appendix in the camera-ready version, if this paper gets accepted.

---

### Author Rebuttal · Authors · 2023-08-08

We thank the reviewers for their valuable comments and suggestions. Here is a summary of some of the key points in the response, including an uploaded PDF that includes additional figures:

- Additional numerical results (in response below and in figures in PDF) of ‘Methods of Line PINNs’ and other PINNs experiments that further support the claim of the paper.

- More detailed discussion of numerical experiments that show that the low-rankness of the Jacobian depends on the model/problem complexity, which is in agreement with the results shown in the paper. Additional plots that demonstrate speedups of 1-3 orders of magnitude of RSNG on all the equations considered.

- A proof-of-concept experiment that indicates that our RSNG approach will also be applicable on higher-dimensional PDE problems, where (dense) Neural Galerkin schemes have shown promising results.

- We further clarify that our goal is to demonstrate that our RSNG approach can overcome two issues (costs, overfitting) that plague (dense) Neural Galerkin methods. Overcoming these issues is a critical step so that Neural Galerkin and related schemes can be successfully applied to classes of problems where traditional numerical solvers struggle, such as for high-dimensional PDEs, nonlinear model reduction and complex geometries (see Conclusions in paper).

We will incorporate the comments and the additional results in the camera-ready version, if accepted. We thank the reviewers again and hope the discussion below and the additional numerical results address the reviewers’ concerns.

---

### Decision · Program_Chairs · 2023-09-21

**Decision:**

Accept (spotlight)

**Comment:**

This is a strong submission. Novelty and relevance to the NeurIPS community have been acknowledged by all reviewers. There were multiple suggestions regarding the positioning and clarity of the submission and we strongly encourage the authors to incorporate the feedback and especially the promised improvements in the final version of the paper.